# Long-term forest-line dynamics in the French Pyrenees: an accelerating upward shift related to forest context, global warming and pastoral abandonment

Noémie Delpouve[1], Laurent Bergès[2], Jean-Luc Dupouey[1], Sandrine Chauchard[1], Nathalie Leroy[1], Erwin Thirion[1], Cyrille B. K. Rathgeber[1]

[1]Université de Lorraine, AgroParisTech, INRAE, SILVA, F-54000 Nancy, France
[2]Université Grenoble Alpes, INRAE, LESSEM, 38402 Saint-Martin-d'Hères, France

*Correspondence to*: Noémie Delpouve (noemie.delpouve@outlook.com)

**Abstract.** Worldwide, the upper forest line has climbed over the past decades, shaping mountain landscapes in response to global changes. In European mountains, this recent trend is a continuation of the forest transition initiated in the mid-19th century, when forest extent was minimal. This study aimed to (1) reconstruct the forest-line dynamics for the entire French Pyrenees from the mid-19th century until today and (2) investigate the influence of human and environmental drivers on the spatio-temporal variations in forest-line shift. To ascertain the forest-line elevational shift for the 114 municipalities studied, three digital land-use maps (dated 1851, 1993 and 2010) were employed. The forest-line shift velocity was calculated for the two periods delineated by these maps. We applied linear mixed-effect models to analyse the role of human and environmental drivers on the forest-line shift. The mean upward shift was 0.9 m.yr$^{-1}$ during the 1851-1993 period but was four-fold higher during the 1993-2010 period (3.5 m.yr$^{-1}$). During the first period, the forest line shifted upward seven times faster in the eastern Pyrenees, where the mountain pine, a pioneer species, formed the ecotone and pastoral abandonment occurred earlier, than in the western Pyrenees (1.3 vs. 0.2 m.yr$^{-1}$). Conversely, in the following period, the shift occurred three times as fast in the western Pyrenees, where abandonment became widespread, as in the eastern Pyrenees (5.6 vs. 2.1 m.yr$^{-1}$). In addition, during the second period, the closed forest line climbed twice as fast as the forest line (5.6 m.yr$^{-1}$), indicating a pronounced densification of the subalpine forest. Our original approach integrates a large spatial scale and temporal depth and sheds new light on the interrelationships between global warming, pastoral abandonment and the forest-line upward shift.

## 1 Introduction

The position of the upper limit of the forest (i.e. the forest line) is a major characteristic of mountain landscapes resulting from complex interactions between climatic factors (heat and water balance), topography, geomorphology, soil conditions, land use, landscape context and the life traits of the tree species forming the forest line (Holtmeier and Broll, 2019; Körner, 2012). Indeed, temperature is a crucial limiting factor in forest-line dynamics, as the isotherm of 6°C during the growing season defines the potential treeline (i.e. upper limit of the tree life form, Körner, 2012; Paulsen and Körner, 2014). In this paper, we

called this limit the *potential forest line*. Furthermore, palaeoecological studies, covering the period of time back to the early Holocene, found downward shifts in the forest line related to the spread of pastures during the Neolithic, indicating that past forest lines were potentially higher than they are currently in the European Alps and Pyrenees (Jalut et al., 1996; Leunda et al., 2019; Schwörer et al., 2014; Van Der Knaap et al., 2012). Moreover, growth, densification and colonisation processes at the forest line are partly determined by the life-history traits of the tree species that form the forest line: seed production, seed

dispersal capacity, seed recruitment, tree growth rate and tree sensitivity to wild animal or livestock herbivory (Cairns and Moen, 2004). These factors interact in different ways over space and time, and drive local forest-line dynamics and the displacement of the adjacent vegetation belts. They also influence the structure and composition of the subalpine forest, in particular in the European mountains where there is a long history of intense human activity (Batllori et al., 2010; Gehrig-Fasel et al., 2007; Palombo et al., 2013).

The middle of the 19[th] century marked the conclusion of the Little Ice Age and the advent of a rural exodus, and since then, global changes (climate and land-use changes) have been affecting forest-line dynamics. Climate change, and particularly current warming trends, may induce important upward shifts in the isotherm - and thus in the forest line (Körner and Hiltbrunner, 2024). Indeed, the air temperature has risen by 1.5°C globally since 1850 and since 1980, each decade has been warmer than the previous one (Gulev et al., 2021). Rural exodus and the decline of traditional pastoralism in European

mountains have also occurred from the early 19[th] century onwards, with an intensification in the middle of the 20[th] century due to the rise of industrial agriculture after the Second World War (Lasanta et al., 2017; MacDonald et al., 2000). In France, the 1850s coincided with the minimum forested area for the territory and marked the beginning of a transition, a switch from major net deforestation to net reforestation (Bergès and Dupouey, 2021; Mather et al., 1999; Rudel, 1998) and potential rise in the altitudinal limit of forests.

Numerous studies report upward shifts in the forest line worldwide since the 20[th] century (Hansson et al., 2021; Harsch et al., 2009; Lu et al., 2021). Forest densification in the alpine ecotone is also frequently reported in the literature, either through tree growth or an increase in recruitment (Améztegui et al., 2010; Bayle et al., 2025; Camarero et al., 2017; Gehrig-Fasel et al., 2007; Sanjuán et al., 2018; Treml and Chuman, 2015; Vitali et al., 2019). At numerous sites throughout Europe, similar upward shifts have been shown in the last 170 years through the study of historical maps and dendrochronology (Hagedorn et al., 2014;

Leonelli et al., 2011; Mainieri et al., 2020; Mietkiewicz et al., 2017; Motta et al., 2006; Shandra et al., 2013; Shiyatov et al., 2007; Tasser et al., 2007; Tattoni et al., 2010). Recent upward shifts in the forest line have also been reported at large spatial scales thanks to aerial photographs (Améztegui et al., 2016; Gehrig-Fasel et al., 2007). However, only a limited response of the forest line, or a lag in the response, has been documented in several locations across Europe despite increasing temperatures (Gehrig-Fasel et al., 2007; Körner and Hiltbrunner, 2024; Paulsen et al., 2000).

A similar rate of warming with an accelerating trend has been observed in the Pyrenees (Moreno et al., 2018). Therefore, a general upward shift in the forest line is expected, corresponding to the reported increase in temperature. In the Pyrenees, densification and infilling processes below the treeline seem to be more frequent than a (true) upward shift in the forest line itself (Batllori et al., 2009; Camarero and Gutiérrez, 2004).

During the 18th and 19th centuries, forests were used for energy production, mostly for industry, and the intensive use of wood
for charcoal led to forest homogenisation in terms of stand structure and tree species composition, favouring for example beech
(*Fagus sylvatica* L.) forests versus silver fir (*Abies alba* Mill.) in the Central Pyrenees (Py-Saragaglia et al., 2017; Saulnier et
al., 2020). Nowadays, forests are used less intensively, and mainly for timber production, but tree species distribution still
reflects legacies of past silvicultural practices. In the French Pyrenees, the eastern forest line is currently dominated by *Pinus
uncinata* Ramond, an early-successional tree species characterised by a high dispersal capacity, rapid growth and low
palatability. In the central and western part, *F. sylvatica* and *A. alba* are often found at the current forest line. They are late-
successional tree species characterised by slow colonisation, relatively slow growth and moderate-to-high palatability. We
expected that variations in tree species composition would be key in modulating the magnitude of the shift in forest line
(Rabasa et al., 2013) and would result in faster upward shifts in the eastern Pyrenees than in the western Pyrenees.

Moreover, the amount of forested area in the Pyrenees has varied in space and over time due to heterogeneous past forest and
land use at the landscape scale. More forest in the surrounding landscape could result in a "forest mass effect", with faster tree
colonisation and forest densification (Abadie et al., 2018a), i.e. a faster upward shift in forest line. We thus hypothesised that
a larger forested area at the beginning of the study period would constitute a more abundant seed source that should facilitate
the upward shift, resulting in a faster change in elevation.

The amount of forested area also depends on the extent of pastureland in the French Pyrenees, where an important tradition of
sheep pastoralism exists and seasonal transhumance occurs from high-elevation pastures to lowland farms in winter
(Rinschede, 1977). The practice of transhumance declined considerably in the Pyrenees, where pastoral abandonment resulted
in an average loss of 75% of the farms between 1945 and 1975 (Métailié, 2006). The pastoral decline was particularly
pronounced in the eastern region. Nevertheless, the transhumant sheep flocks persisted in the western region, where pastoral
areas even increased by 68% between 1972 and 1999 (Eychenne-Niggel, 2003; Rinschede, 1977). In the meantime, in the
easternmost part of the Pyrenees, the earlier pastoral decline continued with a loss of 10% of the remaining pastureland
(Eychenne-Niggel, 2003; Métailié, 2006). The magnitude, temporality and spatial pattern of pastoral abandonment across the
French Pyrenees since the mid-19th century has released the forest edge from grazing pressure to varying degrees. We therefore
expected faster forest-line upward shifts in the eastern Pyrenees than in the western Pyrenees.

Regional long-term trends of forest-line dynamics that integrate the onset of global changes and their recent acceleration are
lacking. Indeed, remote sensing studies do consider large scales but have only been applied to recent forest-line dynamics
assessment - i.e. since the 1950's -, whereas rural exodus and land set aside began much earlier (Lasanta et al., 2017;
MacDonald et al., 2000). Palaeoecology, dendrochronology and historical maps have been widely used to assess forest-line
dynamics over long temporal scales but their use has generally been limited to small areas. Nevertheless, across Europe,
historical maps can cover large areas, as the État-Major map (Kaim et al., 2016). However, the use of historical maps was
usually restricted to small areas (eg. Egarter Vigl et al., 2016; Mainieri et al., 2020; Mietkiewicz et al., 2017; Tasser et al.,
2007) while anthropogenic pressure and forest context vary greatly within any given mountain range, which can limit the
generalisation of such small-size studies to the entire mountain range.

The recent digitisation of the French land-use map dating back to the 1850s and including forests provides the opportunity to use a historical map at the regional scale, in order to go further back in time and to cover a large area. The focus of our research was on the French Pyrenees, as this was the first mountain range to be digitised.

The main objective of our study was to document large-scale patterns of forest-line dynamics and identify common processes and potential heterogeneity across an entire massif. Two subsequent, more specific, goals were: (1) to reconstruct forest-line dynamics in the French Pyrenees from the minimum forest extent in 1850 to the present day, and (2) to relate forest-line dynamics to potential biophysical and anthropogenic drivers. To reach these objectives, we compared three land cover maps in the French Pyrenees mountain range: the Napoleonic military map (known as the État-Major map) produced in 1851, the BD Forêt® v1 (1993) and the BD Forêt® v2 (2010); the latter two maps were provided by the French National Institute for Geographic and Forestry Information (IGN).

## 2 Materials & Methods

### 2.1 Study area

The Pyrenees range stretches over 300 km between the Atlantic Ocean and the Mediterranean Sea, at the French-Spanish border, and is almost 100 km wide in its central part. The maximum elevation is 3404 m a.s.l. at Aneto Peak (Spain), while the highest summit in France is the Vignemale (3298 m a.s.l.). The Pyrenees range exhibits a strong longitudinal climatic gradient. The eastern region is under Mediterranean influence and is characterised by lower precipitation and a warmer average temperature than the western region, under oceanic influence: 1060 vs 2298 mm of average annual precipitation and 5.9 vs 5.3°C between 1958 and 2008, respectively for Cerdagne (eastern French region) and the Pays-Basque massifs (western French region) (Maris et al., 2009). Overall, 59% of the Pyrenees are covered in forests, mainly composed of beech (*Fagus sylvatica* L.), fir (*Abies alba* Mill.) and Scots pine (*Pinus sylvestris* L.) in the montane belt, and dominated by mountain pine (*Pinus uncinata* Ramond) in the subalpine belt (Ninot et al., 2017).

### 2.2 Selection of the studied municipalities

We studied forest-line dynamics at the municipality level in the French part of the Pyrenees range (Fig. 1). Since the forest line has been historically located around 2100-2300 m a.s.l. in the Pyrenees (Feuillet et al., 2020), we only selected municipalities whose maximum elevation exceeds 2200 m a.s.l. After removing the municipalities where no forest was present in 1851 and those where forest had already reached the maximum elevation in the municipality in 1993 or 2010, to avoid calculation bias, the study area finally encompassed 114 municipalities (Fig. 1). The surface area of the municipalities ranged from 461 to 24,864 ha, with an average of 4,419 ha and a total area of 5,038 km². As the valleys in the French Pyrenees generally face north, eastern and western exposures were dominant in our study area.

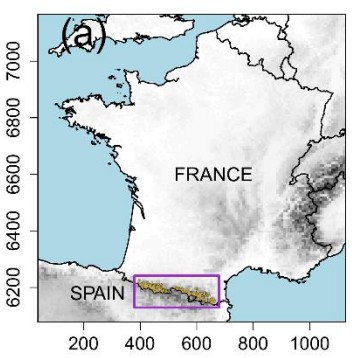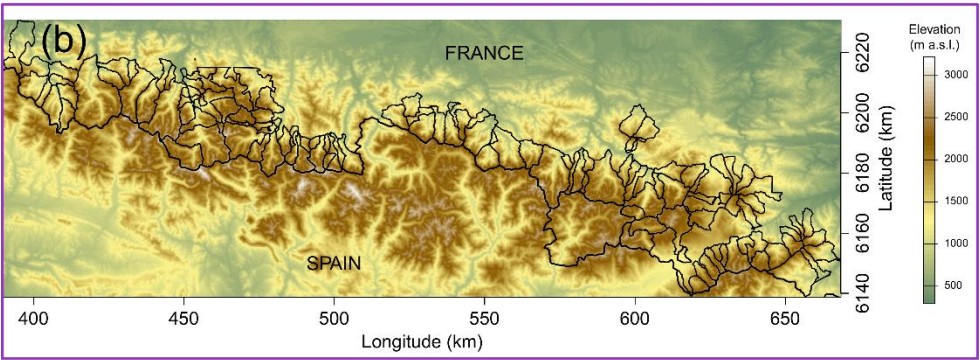

**Figure 1: (a) Location of the study zone in the French Pyrenees. (b) Boundaries (thin lines) and elevation patterns for the 114 municipalities studied along the French-Spanish border (thick line). Coordinates are expressed in the projected RGF93 system. © EuroGeographics provided the digital elevation model for Europe. European Environment Agency provided the countries boundaries. Map lines delineate study areas and do not necessarily depict accepted national boundaries.**

### 2.3 Map descriptions and standardisation

We used three digitised land-use maps covering the French Pyrenees to reconstruct forest-line dynamics from the middle of the 19th century until today. The oldest map is the *État-Major* map (EMM), produced between 1849 and 1854 (with a mean date of 1851) in the study area, at the scale of 1:40,000. The EMM is a military land-use map based on the Napoleonic land registry, covering the whole France. It represents all forest patches of at least 0.1 ha in area (IGN, 2021). The second forest map is the *BD Forêt® v1* (BDF1), which is based on aerial photographs, at scales ranging from 1:17,000 to 1:25,000, taken between 1987 and 1999 (with a mean date of 1993) in the study area (IGN, 2018). The third forest map, *BD Forêt® v2* (BDF2), is based on aerial photographs taken between 2006 and 2015 (with a mean date of 2010) in the study area (IGN, 2016). The two *BD Forêt®* maps include forest stand types within forest patches of at least 2.25 and 0.5 ha, respectively for BDF1 and BDF2. These three maps only were available in a vectorised version at the scale of the entire French Pyrenees massif when we started this study.

To make the three maps (EMM, BDF1 and BDF2) comparable, we standardised them in six steps. (1) We cropped the EMM, BDF1 and BDF2 according to the boundaries of the study area, and added a temporary 1 km buffer zone at the edge of the study area to include forest patches of which only a part was within the study area, but which belong to larger forest patches extending beyond the study area. We could not find a suitable forest map in Spain dating from the 19th century (Ruiz Del Castillo Y Navascues et al., 2006), and, in most cases, the boundary follows the ridge, therefore, we did not add the 1 km buffer in foreign countries. (2) As BDF1 and BDF2 distinguish between two types of forest, open (canopy cover between 10 and 40%) and closed (canopy cover above 40%), we aggregated the open and closed forest polygons to estimate the overall forest line, and considered only closed forest polygons to estimate the closed forest line. (3) In each map, to compensate for the wider-than-real road layout in the EMM and the non-vectorization of forests less than 75 m wide in BDF1 (IGN, 2018),

we aggregated the polygons spaced less than 75 m apart. (4) In each map, the surface area of each forest patch was calculated. (5) As polygons smaller than 2.25 ha were not vectorised in BDF1 (IGN, 2018), we removed the polygons with a surface area below 2.25 ha from the EMM and BDF2. (6) Finally, the buffer zone was removed and the forest polygons were subdivided by municipality. In this way, the forest information on each map was aligned with the administrative information for each municipality to which the forest polygons belonged (e.g. name, code), making a direct comparison among the tree maps possible.

Subsets from BDF1 and BDF2 containing only closed forest patches were created and treated following the same steps. Therefore, after the standardisation process, we finally obtained five distinct forest vector maps: one for the EMM and two each for BDF1 and BDF2, one for all forests and one for closed forests only.

## 2.4 Estimating the shift in forest line

We developed an original method combining the forest maps with a digital elevation model to estimate the forest-line elevation for each municipality (Fig. 2). The digital elevation model was provided by the IGN and has a spatial resolution of 25 m with elevation values rounded to the nearest metre (IGN, 2017). For each municipality in the study area, we rasterised the forest present in each vector map (Fig. 2a). The resulting raster presented a spatial resolution of 25 m and contained the fraction of each pixel that was covered by the forest polygons. We combined the rasters from the digital elevation model and the five rasterised forest maps, delivering a table of dimensions n x 6, with n the number of pixels in the municipality and 6 the number of columns corresponding to each raster. To estimate the distribution of forest cover according to elevation, we used the raster table to calculate the mean elevation and the percentage of forest cover inside incremental bands of 100 m in elevation (Fig. 2b). Then, starting from the top, we slid this moving elevation band downwards, metre by metre, to cover the range of elevations present in the municipality. Of the 25 m, 50 m and 100 m elevational bands that we tested, the 100 m band showed the most consistent distribution of forest cover according to elevation with the least background noise (Fig. 2). It should be noted that the area of land contained in the band varied with elevation, increasing as the band was moved downwards, while the elevation range remained fixed at 100 m. Next, we adjusted a general additive model (GAM) to the percentages of forest cover with the mean elevation of the pixels within the band as the predictor variable (Fig. 2c). Finally, starting from the top, we searched downwards for the first elevation at which the percentage of forest cover exceeded 5%. This elevation defined the forest line in the municipality. We also tested a threshold of forest cover of 2.5%, and no threshold, to assess forest-line elevation but finally did not keep it as we found too many small wooded tracts were retained below the forest line (potential mapping errors).

We computed shifts in the forest line as their difference in elevation between 1851 and 1993, between 1993 and 2010 and between 1851 and 2010 (Fig. 2c). The closed forest-line shift was computed between 1993 and 2010. In addition, we also computed the velocity of forest-line and closed forest-line shifts (in m per year) for each municipality, based on the precise year of mapping for greater accuracy.

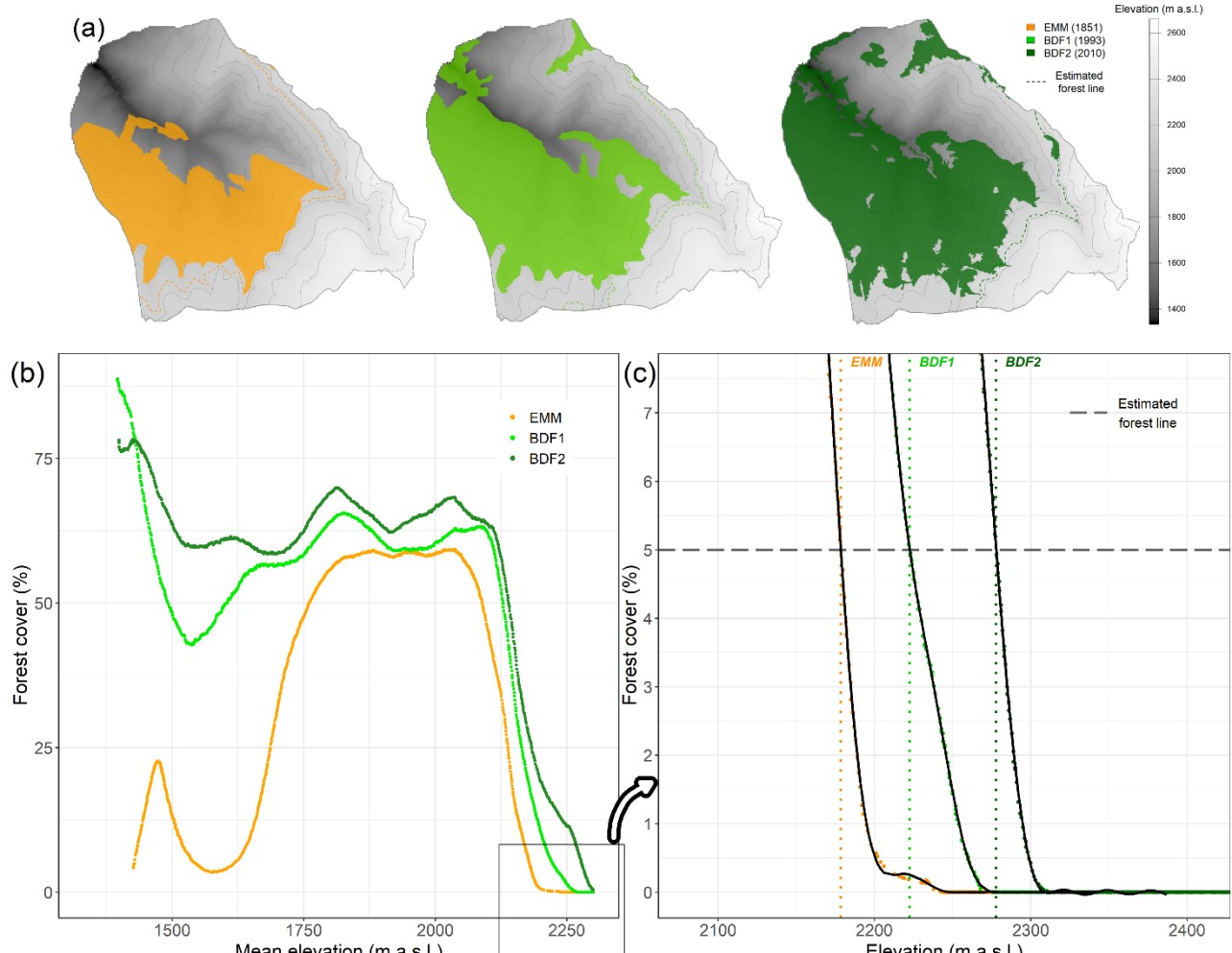

**Figure 2: Estimation of the elevation of the subalpine forest line (all forests) and shifts in the forest line for the municipality of Valcebollère (66220) on the *État-Major Map* (EMM), and in *BD Forêt® v1* (BDF1) and *BD Forêt® v2* (BDF2). (a) Forest cover on the three maps. (b) Percentage of forest cover as a function of mean elevation in a moving band of 100 m elevation range. (c) Focus on the part of the distribution where the curves cross the threshold of 5% forest cover. A GAM model (in black) was fitted to the curves. The elevation at which the GAM model crossed the 5% threshold gave the forest-line elevation for each map. Forest-line shifts were calculated as the difference between two consecutive maps.**

## 2.5 Drivers of forest-line dynamics

To investigate the factors explaining the spatio-temporal dynamics of the forest line, we collected additional topographic, climatic, landscape, forest and socio-economic data, either for the whole study area (Fig. 3) or at the municipality scale (Table 1). These data were used in two different analyses: (1) to compare the temporal trends of forest-line dynamics and its potential drivers, and (2) to assess the effects of the factors driving spatial variations.

## 2.5.1 Topographic and climatic drivers

Firstly, we assessed the extent of global warming in the study area. Indeed, we calculated a moving average over 30 years based on the mean temperatures at the Pic-du-Midi weather station (2877 m a.s.l.) from 1881 to 2019, provided by Météo-France. Only data from the Pic-du-Midi weather station allowed us to go back to the 19th century to assess temporal variations in temperature. The other weather stations in the Pyrenees had shorter time series and were located at elevations further lower than the forest line.

Secondly, we focused on recent spatialised data to assess climatic variations across the study area. To account for spatial variations in temperature and water availability, both of which affect tree growth and survival, mean temperature in the warmest month of summer (June to August), mean temperature in the coldest month of winter (December to February) and total annual precipitation (sum of the monthly mean precipitation) were calculated for the period 1981-2010 from the Aurelhy database (Bénichou and Le Breton, 1987). We also calculated the mean summer water balance extracted for the period 1961-1990 from the DIGITALIS database (UMR SILVA: Université de Lorraine-AgroParisTech-INRAE; https://silvae.agroparistech.fr/home/; Bertrand et al., 2011; Piedallu and Gégout, 2007) to characterise the amount of water available during the growing season. We calculated the mean annual radiation between 1971 and 2000 extracted from the Hélios radiation model at a 1 km resolution (DIGITALIS database; Piedallu and Gégout, 2008, 2007) to estimate the amount of light available for tree photosynthesis. To address the effect of summer drought, we also calculated the mean summer aridity index extracted from the Global Aridity Index and Potential Evapo-Transpiration (ET0) Database (https://csidotinfo.wordpress.com/data/global-aridity-and-pet-database/; Zomer et al., 2022). We calculated the variables mentioned above in the area of the municipality located 300 m above the forest line in 1851 and 1993, and above the closed forest line in 1993, to account for the spatial variations in climatic conditions among municipalities, while including an area small enough to be representative of conditions where forest-line shifts should occur, but large enough to include enough points for a meaningful average.

We also calculated the mean elevation of the municipality to characterise the average topography of the municipality. In addition, we calculated the mean slope and northness (exposure cosine) in the 300 m above the forest or closed forest line with the digital elevation model (IGN, 2017), since these parameters may affect forest colonisation and management.

**Table 1: Summary of the variables tested in the linear mixed models. The continuous variables specific to each model that were calculated based on forest-line elevation in 1851, and forest-line and closed forest-line elevations in 1993 are presented in (a). Continuous and discrete variables independent of forest-line elevation, used in several models, are presented in (b). The expected effect of each variable on the forest-line shift velocity is given, with + meaning a faster upward shift and – a slower upward shift. Cells with grey text indicate variables that were excluded from subsequent analyses in order to avoid correlations greater than 0.7. The dominant tree species class was excluded for the period 1993–2010.**

(a)

| Predictor variable | Source | Unit | Expected effect on forest-line shift | Mean and standard deviation (Sd) | | |
|---|---|---|---|---|---|---|
| | | | | Forest 1851 | Forest 1993 | Closed forest 1993 |
| Mean annual radiation in the 300 m above the line | Digitalis | $Kj.cm^{-2}$ | + | Mean: 38 Sd: 4.0 | Mean: 38 Sd: 4.1 | Mean: 38 Sd: 4.1 |
| Mean temperature in the warmest month of summer in the 300 m above the line | Aurelhy | °C | + | Mean: 20 Sd: 1.4 | Mean: 19 Sd: 1.2 | Mean: 20 Sd: 1.2 |
| Mean temperature in the coldest month of winter in the 300 m above the line | Aurelhy | °C | + | Mean: -4.2 Sd: 1.3 | Mean: -4.5 Sd: 1.3 | Mean: -4.2 Sd: 1.3 |
| Total annual precipitation in the 300 m above the line | Aurelhy | mm | + | Mean: 1322 Sd: 235 | Mean: 1343 Sd: 224 | Mean: 1324 Sd: 221 |
| Mean summer water balance in the 300 m above the line | Digitalis | mm | + | Mean: -4.7 Sd: 19 | Mean: -2.2 Sd: 14 | Mean: -5.1 Sd: 14 |
| Mean summer aridity index in the 300 m above the line | Global-AI_PET_v3 | - | + | Mean: 0.59 Sd: 0.05 | Mean: 0.60 Sd: 0.04 | Mean: 0.59 Sd: 0.04 |
| Mean slope in the 300 m above the line | Digital elevation model | ° | + | Mean: 27 Sd: 5.5 | Mean: 27 Sd: 5.7 | Mean: 27 Sd: 5.8 |
| Mean northness in the 300 m above the line | Digital elevation model | - | - | Mean: 0.16 Sd: 0.28 | Mean: 0.18 Sd: 0.29 | Mean: 0.16 Sd: 0.30 |
| Proportion of forested area below the line | Forest maps | - | + | Mean: 0.32 Sd: 0.15 | Mean: 0.58 Sd: 0.17 | Mean: 0.48 Sd: 0.19 |
| Forest-line elevation | Forest maps | m a.s.l. | - | Mean: 1884 Sd: 284 | Mean: 2007 Sd: 282 | Mean: 1893 Sd: 288 |
| Surface area per farmer above the line | Forest maps / EHESS / Agreste | $ha.hab^{-1}$ | + | Mean: 2.9 Sd: 3.9 | Mean: 68 Sd: 105 | Mean: 96 Sd: 190 |

(b)

| Predictor variable | Source | Unit | Type | Expected effect on forest-line shift | Mean and standard deviation (Sd) or sample size (N) |
|---|---|---|---|---|---|
| Mean elevation in the municipality | Digital elevation model | m a.s.l. | Continuous | - | Mean: 1657 Sd: 283 |
| Dominant tree species class | *BD Forêt® v2* | - | Discrete | -<br>-<br>-<br>-<br>+<br>+ | Beech (13)<br>Fir (17)<br>Mixed (7)<br>Deciduous (13)<br>Mountain pine (49)<br>Conifers (15) |
| Closed forest dominant tree species class | *BD Forêt® v2* | - | Discrete | -<br>-<br>-<br>-<br>+ | Beech (16)<br>Fir (21)<br>Mixed (18)<br>Deciduous (8)<br>Mountain pine (51) |
| Change in livestock density between 1852 and 2000 | Pastoral survey of 1852 (Demonet, 1990) / Agreste | animals.km$^{-2}$ | Continuous | - | Mean: -75 Sd: 33 |
| Change in livestock density between 2000 and 2010 | Agreste | animals.km$^{-2}$ | Continuous | - | Mean: -3 Sd: 1 |

### 2.5.2 Landscape and forest context drivers

Firstly, to compare temporal forest-line dynamics and overall forest expansion, the total forest area in the study region was calculated for the three forest maps (EMM, BDF1, BDF2) and completed with the forest area present in 1908 according to the Daubrée forest inventory (Daubree, 1912). This forest inventory only reported forest area figures and did not provide any precise spatial information.

Secondly, we identified the dominant tree species class at the forest line, defined as the most frequent tree species class in the

235 forest pixels on BDF2 above the estimated forest-line elevation to determine the forest context at the municipality scale (Table S1). BDF2 contains six tree species categories: "Conifers", "Mountain pine", "Fir", "Mixed", "Deciduous" and "Beech" at the forest line (Table S1, Fig. S2a). The two classes "conifers" and "deciduous" correspond to undifferentiated coniferous and deciduous species (i.e. photo-interpretation failed to precisely determine tree species composition). In contrast to the limited number of pixels at the forest line, using pixels above the 5% threshold for forest cover made it possible to integrate a sufficient

number of pixels to be representative of tree species composition at the forest line in each municipality. We applied the same

methodology to determine dominant tree species class at the closed forest line. To avoid classes with too small a sample size, we grouped the initial six tree species categories in BDF2 into five classes based on tree species identity and ecology: "Mountain pine", "Deciduous", "Mixed", "Fir" and "Beech" (Table S1, Fig. S2b). Based on the dominant tree species class found at the closed forest line, we defined a more precise dominant tree species class at the forest line as follows: when the dominant tree species class at the forest line was "Conifers" or "Deciduous" and the corresponding tree species class at the closed forest line was "Mountain pine", "Fir" or "Beech", we replaced the tree species class attributed at the forest line by the tree species class at the closed forest line (Table S1, Fig. S2c). Otherwise, the class originally affected at the forest line was kept.

Moreover, for each municipality, we calculated the proportion of forested area below the forest line for the EMM, and below the forest line and closed forest line for BDF1.

### 2.5.3 Socio-economic drivers

Firstly, to reveal the temporal dynamics of the rural exodus, we collected the human population from the oldest census in 1793 to the one in 2019 (provided by EHESS (http://cassini.ehess.fr) and INSEE (https://www.insee.fr)) and summed it by year across the entire study area. We also collected the number of sheep and cattle in 1838, 1852, 2000 and 2010, from pastoral surveys and the Agreste website at the "arrondissement" (i.e. groups of municipalities) scale (https://agreste.agriculture.gouv.fr; Demonet, 1990; Goüin, 1840), then calculated the livestock density in the "arrondissements" in the study area for each group, to provide insight into the past and present variations in livestock that could potentially influence the observed temporal trend in forest colonisation.

Secondly, to assess spatio-temporal variations in socio-economic factors, and relate these to spatial variations in forest-line upward shift, we calculated the change in total livestock (cattle, sheep, goats, equines and pigs) density between 1852 and 2000 (livestock density in 2000 - density in 1852), and between 2000 and 2010 (livestock density in 2010 - density in 2000) from the pastoral surveys cited above, assigning the value of the arrondissement to the municipalities it included. Moreover, we calculated the area above the forest line and divided it by the number of farmers in 1851 and in 1993 to estimate the mean area available for each farmer. We assumed that high values corresponded to less intensive management. For 1851, we generally used the number of inhabitants in the municipality, which we considered to be close to the number of farmers at the time. For 1993, we used the number of permanent farm workers in the municipality in 1988, available from the Agreste website.

### 2.6 Statistical analysis

We applied mixed linear models at the municipality scale (1) to test whether upward shifts in forest line (between 1851 and 1993, between 1993 and 2010 and between 1851 and 2010) were significantly different from 0; (2) to test whether the velocity of shift between the two periods (1851-1993 vs 1993-2010) was significantly different; (3) to test whether the velocity of shift was significantly different for the forest line and for the closed forest line; and (4) to compare the magnitude of the shifts between the western and eastern regions of the Pyrenees range. We included the variable "arrondissement" as a random factor

in our models to account for the spatial structure of our sampling scheme. The first three models were built using only the "arrondissement" as a random effect, with no fixed effects, to test if the differences were significant, i.e. similar to t-tests but accounting for spatial autocorrelation. We systematically applied the Moran I test to check for spatial autocorrelation in model residuals; none were detected (Moran, 1950). For the last test (4) the forest-line shift velocity between 1851 and 1993 and between 1993 and 2010 were the response variables and the effect of the spatial group was tested (west vs east, separated at an RGF93 longitude of 520 km).

To assess the role of the potential drivers on the velocity of the shifts in forest line at the municipality scale, we fitted three multiple linear models: two for the forest line between 1851 and 1993, and between 1993 and 2010, and one for the closed forest line between 1993 and 2010. For each model, we started with the 15 variables cited above and summarised in Table 1. To avoid statistical problems, we checked that the correlations among the variables were below 0.7 (Zuur et al., 2010). For pairs of variables with correlations above 0.7, we removed the one that was the most correlated with other variables (Table 1, Fig. S3). One outlier municipality was removed from the analysis of the variations in the velocity of shift in the closed forest line (Viey, Table S4).

Comparing all possible models using the three sets of variables identified above, we selected the models with *delta(AICc)<2* compared to the model with the lowest *AICc*. Among these models, the one with the minimum number of variables was selected as the most parsimonious. The final selected model explaining variations in (1) forest-line shift velocity between 1851 and 1993 included mean summer water balance and mean slope in the 300 m above the forest line in 1851, dominant tree species class, forest-line elevation in 1851, and surface area per farmer above the forest line in 1851; (2) forest-line shift velocity between 1993 and 2010 included mean summer aridity index in the 300 m above the 1993 forest line, and change in livestock density between 2000 and 2010; (3) closed forest-line shift velocity between 1993 and 2010 included mean summer water balance and mean slope in the 300 m above the 1993 closed forest line, proportion of forested area below the closed forest line, and closed forest-line elevation in 1993. Finally, to compare effect magnitude, we calculated the R² contribution of each variable by removing it from the most parsimonious model.

### 2.7 Software

Map standardisation was performed with the ArcGis Pro software version 3.0.2. Data were prepared and analysed with R version 4.3.3. Forest-line elevations were estimated thanks to a dedicated script that based on the *sf* (Pebesma, 2018), *terra* (Hijmans et al., 2024) and *mgcv* (Wood, 2011) packages. Spatial autocorrelation was assessed with the R package *ape* (Paradis and Schliep, 2019). Finally the models were built with the *lme* function of the R package *nlme* (Pinheiro et al., 1999), and the parsimonious models were determined with the *dredge* function of the *MuMIn* package (Bartoń, 2023). The graphics were created using the *ggplot2* package (Wickham, 2016).

## 3 Results

### 3.1 Accelerated temperature increase, comparable forest expansion and massive pastoral abandonment

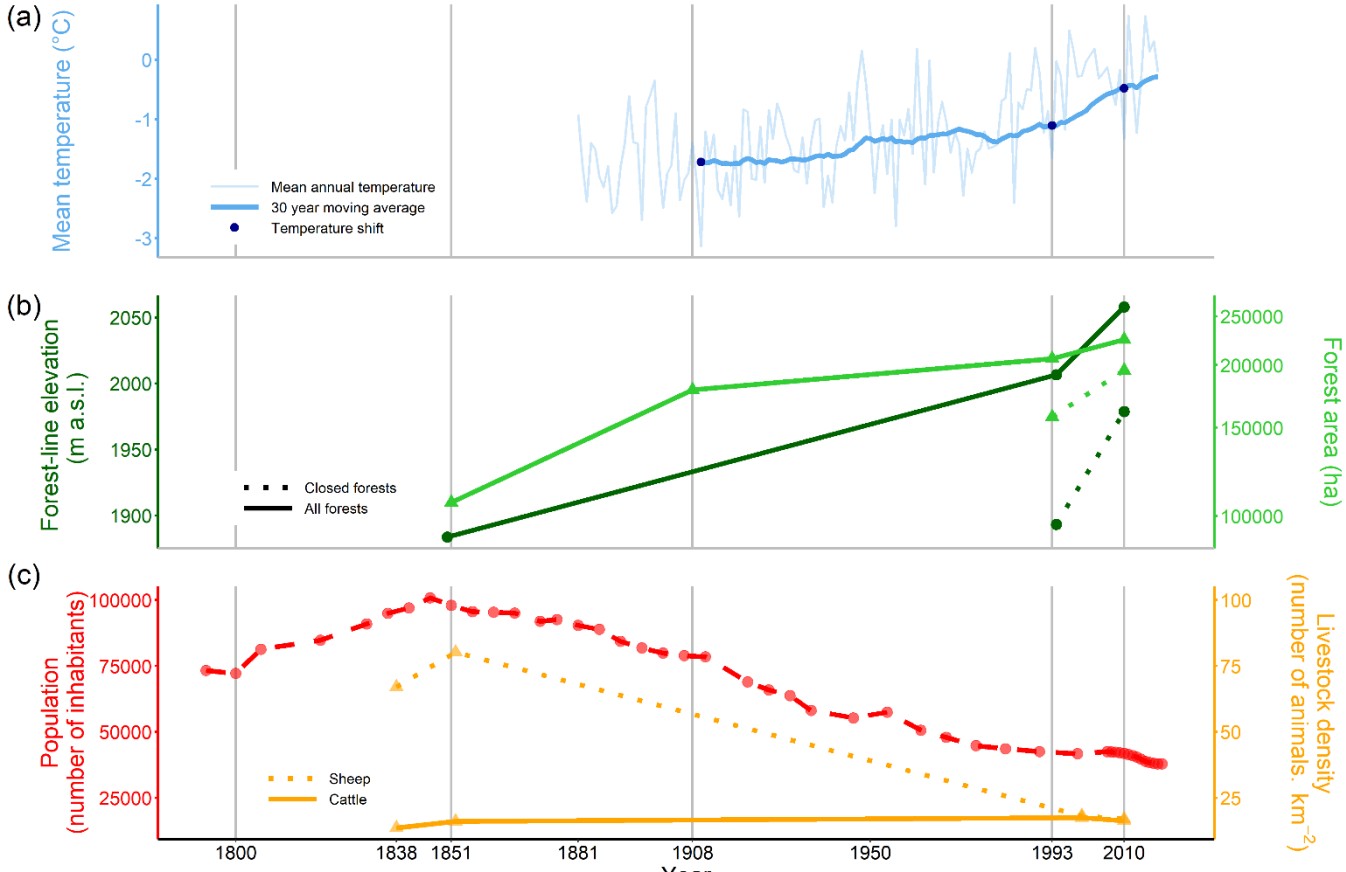

Figure 3: Change in forest-line elevations and potential climatic and socio-economic drivers at the scale of the French Pyrenees during the 19th and 20th centuries. (a) Change in the annual mean temperature at the Pic-du-Midi weather station (2877 m a.s.l.). The 30-year moving average was calculated with the mean annual temperature from 1881 to 2019 (Météo-France). The temperature shift was estimated for the studied period. (b) Changes in the elevation (●) of the forest line and the closed forest line and in the forest area (▲) in the studied municipalities, based on the *État-Major* map (1851), *BD Forêt® v1* (1993) and *BD Forêt® v2* (2010), and the Daubrée inventory (1908). Note that the Daubrée inventory only provided figures of forest area without mapping, preventing us to use it to assess forest-line elevation in 1908. The right axis for forest area had been log-transformed. (c) Changes in socio-economic factors. The total population in the study area was obtained from the censuses available from EHESS and INSEE. The change in cattle and sheep density (over 9 "arrondissements") was obtained from archived pastoral surveys and from the Agreste website.

The 30-year moving mean annual temperature at the Pic-du-Midi weather station was -1.72°C in 1910, -1.11°C in 1993 and -0.48°C in 2010. Thus, the temperature rose by 0.62°C between 1910 and 1993, by 0.62°C between 1993 and 2010 and by 1.24°C between 1910 and 2010 (Fig. 3a).

Forest area increased by 1,262 ha.yr$^{-1}$ between 1851 and 1908 (+67%), then slowed to 324 ha.yr$^{-1}$ between 1908 and 1993 (+15%) and then re-accelerated to 1,100 ha.yr$^{-1}$ between 1993 and 2010 (+9%) (Fig. 3b). Closed forest area increased twice as fast as did forest between 1993 and 2010, by 2,200 ha.yr$^{-1}$ (+24%).

The human population in the study area increased by 36% between 1800 and 1851, then decreased by 58% between 1851 and 2010 (Fig. 3c). The same trends were observed for sheep density, which increased slightly between 1838 and 1852, then decreased by 79% between 1852 and 2010, while the number of cattle slightly increased (1%) (Fig. 3c). However, livestock density declined most strongly in the east between 1852 and 2000, while it remained stable in the westernmost part of the study area. Then, between 2000 and 2010, livestock density decreased at the same pace on average throughout the zone (Fig. S5).

### 3.2 Spatio-temporal dynamics of forest-line and closed forest-line elevations

Forest-line elevation averaged 1884 m a.s.l. (± 27) in 1851, 2007 m a.s.l. (± 26) in 1993 and 2058 m a.s.l. (± 26) in 2010 (mean ± SE) (Fig. 4). Thus, the forest line moved upwards by an average of 174 ± 17 m between 1851 and 2010 (t = 4.82, p < 0.001), of which 123 ± 17 m occurred between 1851 and 1993 (t = 2.76, p = 0.007) and 51 ± 8 m between 1993 and 2010 (t = 6.41, p < 0.001). Overall, 88% of the municipalities displayed an upward shift between 1851 and 2010 (82% between 1851 and 1993, and 77% between 1993 and 2010) (Table S4). Moreover, 61% of the municipalities showed a faster upward shift of the forest line after 1993. Indeed, the upward shift was on average four times as fast between 1993 and 2010 as it was between 1851 and 1993 (3.5 ± 0.5 vs. 0.9 ± 0.1 m.yr$^{-1}$, t = 2.75, p = 0.007; Fig. 5).

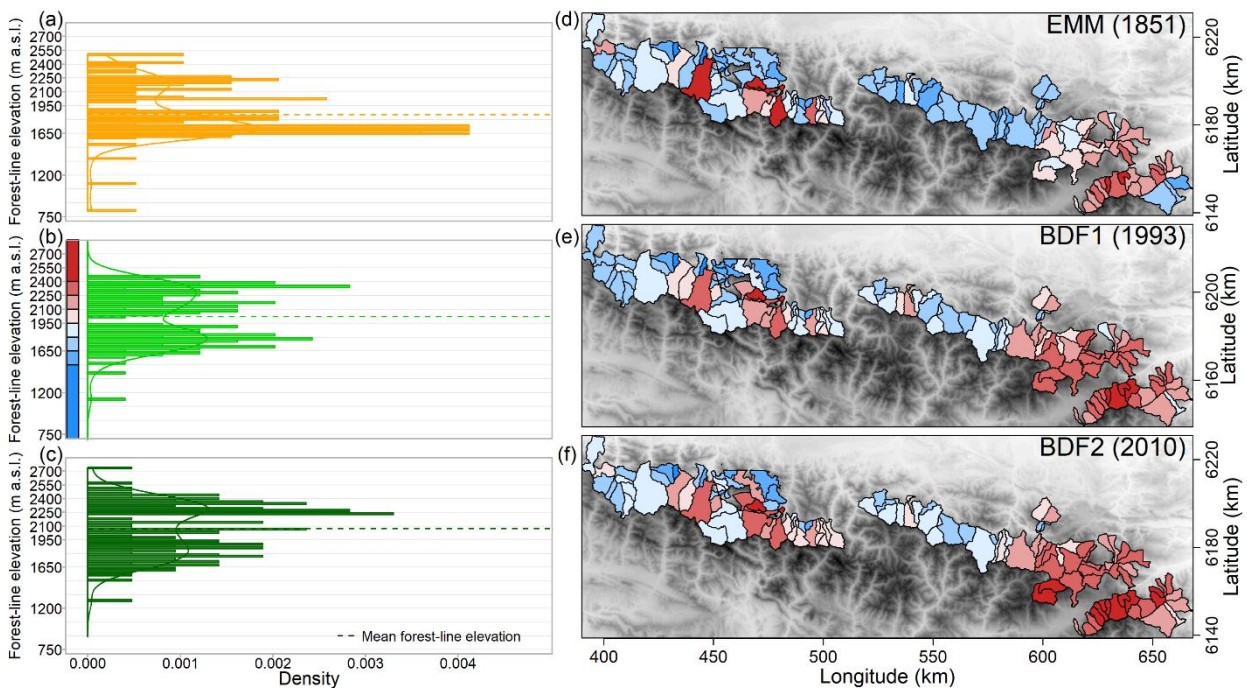

**Figure 4: Distribution of forest-line elevations in the 114 studied municipalities in the French Pyrenees on (a) the *État-Major* map (EMM), (b) *BD Forêt® v1* (BDF1) and (c) *BD Forêt® v2* (BDF2). Spatial distribution of forest-line elevations in each map, successively (d) EMM, (e) BDF1 and (f) BDF2. The palette of colours in d-f corresponds to the elevation scale shown in (b). © EuroGeographics provided the digital elevation model for Europe.**

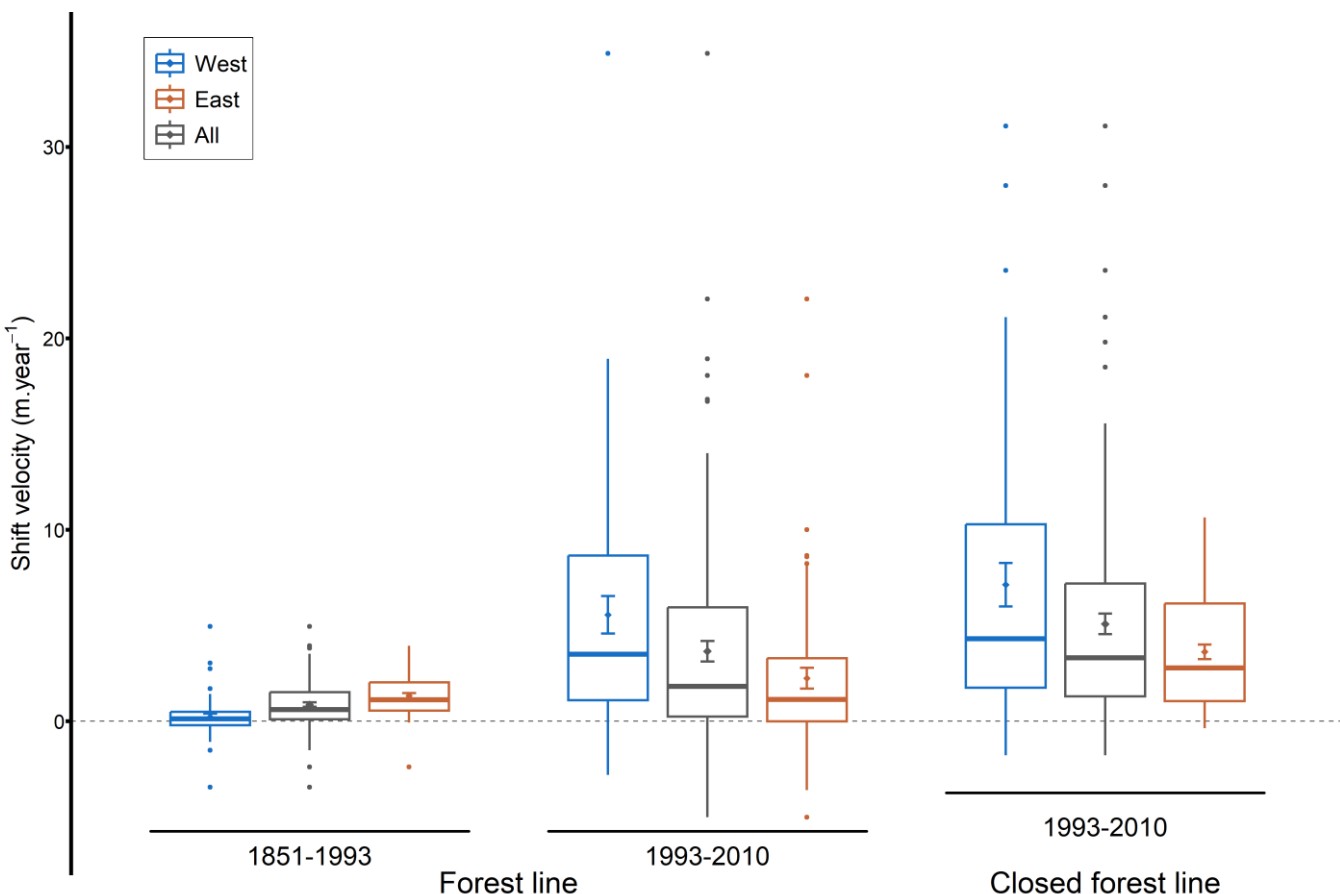

Figure 5: Mean, standard error and boxplot for forest-line and closed forest-line shift velocities in the French Pyrenees for the 1851-1993 and 1993-2010 periods, for the entire study area, and for the western and eastern groups (separated at an RGF93 longitude of 520 km).

Higher forest-line elevations were found in the easternmost part (Catalan Pyrenees) on all three maps (Fig. 4). To the west, several municipalities also displayed forest lines at high elevations, although the elevations were generally lower. Between 1851 and 1993, the forest-line shift was faster in the eastern than in the western Pyrenees, with the mountain range divided at an RGF93 longitude of 520 km ($1.3 \pm 0.1$ m.yr$^{-1}$ in the east vs. $0.2 \pm 0.2$ m.yr$^{-1}$ in the west, F = 24.82, p < 0.001; Fig. 5, 6a). On the contrary, between 1993 and 2010, the upward shift was slower in the east than in the west ($2.1 \pm 0.6$ vs. $5.6 \pm 1.0$ m.yr$^{-1}$, F = 10.62, p = 0.001; Fig. 5, 6b).

The closed forest-line elevation averaged 1893 m a.s.l. ($\pm 27$) in 1993 and 1979 m a.s.l. ($\pm 25$) in 2010. Consequently, the closed forest line shifted upward by $85 \pm 9$ m on average between 1993 and 2010 (t = 6.23, p < 0.001), corresponding to a velocity of $5.6 \pm 0.7$ m.yr$^{-1}$. This upward shift in closed forest line occurred in 93% of the municipalities. On average, the closed forest-line shift was faster than the forest-line shift (t = -2.00, p = 0.048, Fig. 5) in the majority of municipalities (61%). As for the forest line, the upward shift in the closed forest line between 1993 and 2010 was slower in the east than in the west ($3.6 \pm 0.4$ vs. $8.3 \pm 1.6$, F = 10.58, p = 0.002; Fig. 5, 6c).

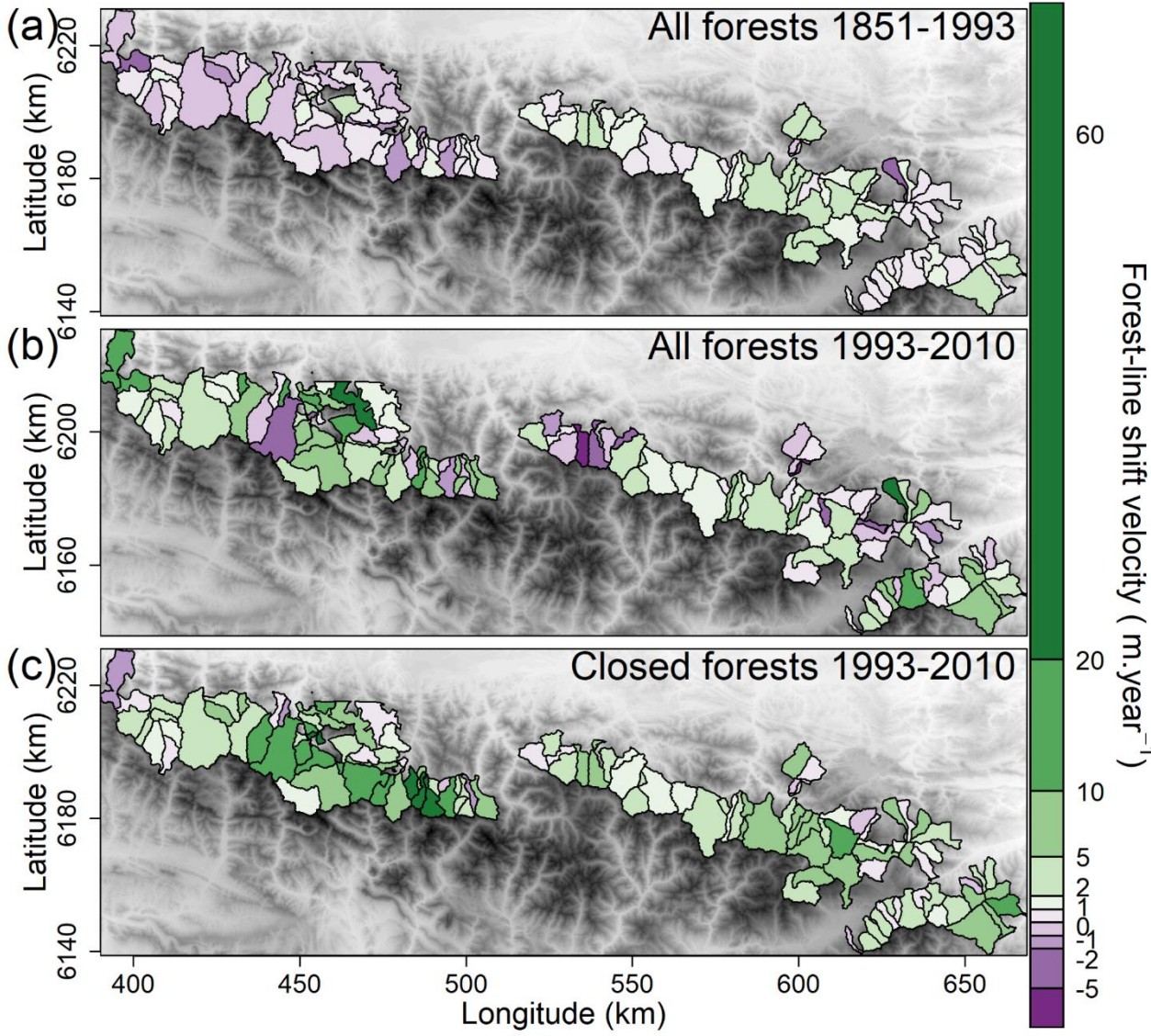

**Figure 6: Spatial distribution of (a) forest-line shift velocities for the 1851-1993 period, (b) forest-line shift velocities for the 1993-2010 period and (c) closed forest-line shift velocities for the 1993-2010 period, in the French Pyrenees. The classes of the shift velocity scale were manually determined. © EuroGeographics provided the digital elevation model for Europe.**

### 3.3 Spatio-temporal drivers of forest-line and closed forest-line dynamics

The most parsimonious model explained 67% of the total variation in the velocity of the forest-line shift between 1851 and 1993 and included five variables. Dominant tree species class was the highest contributing factor and the forest line shifted

upward the fastest between 1851 and 1993 when the forest line was composed predominantly of mountain pine, followed by conifers, deciduous trees and fir, but remained virtually stable when the forest line was composed of mixed-species stands or beech. Furthermore, the forest-line shift for the 1851-1993 period was faster in municipalities exhibiting, by order of importance: (1) a lower forest-line elevation in 1851, (2) a lower mean summer water balance, (3) a higher mean slope, and (4) a larger area per farmer above the forest line (Table 2).

**Table 2: Estimates with associated standard errors, t-values and p-values for the linear model predicting forest-line shift velocity between 1851 and 1993 in the French Pyrenees based on mean summer water balance and mean slope in the 300 m above the 1851 forest line, dominant tree species class, forest-line elevation in 1851, and surface area per farmer above the forest line in 1851. F-values and associated p-values of the ANOVA for this linear model are presented for the discrete variable 'dominant tree species class'. The dominant tree species class "Beech" corresponds to the intercept. The contribution of each variable to the total adjusted $R^2$ of 0.67 is presented in the last column.**

|  | Estimate | Std. Error | t value | p value | $R^2$ contribution |
|---|---|---|---|---|---|
| Total |  |  |  |  | 0.67 |
| Intercept | 3.44 | 0.67 | 5.12 | < 0.001 | - |
| Mean summer water balance | -0.02 | $4.26e^{-3}$ | -4.32 | < 0.001 | 0.06 |
| Mean slope | 0.06 | 0.01 | 4.04 | < 0.001 | 0.05 |
| Dominant tree species class |  | F = 22.57 |  | < 0.001 | 0.33 |
| *Dominant tree species class: Fir* | 0.20 | 0.30 | 0.68 | 0.500 | - |
| *Dominant tree species class: Mixed* | 0.10 | 0.36 | -0.28 | 0.777 | - |
| *Dominant tree species class: Deciduous* | 0.55 | 0.30 | 1.83 | 0.071 | - |
| *Dominant tree species class: Mountain pine* | 2.197 | 0.30 | 7.42 | < 0.001 | - |
| *Dominant tree species class: Conifers* | 0.543 | 0.30 | 1.85 | 0.068 | - |
| Forest-line elevation in 1851 | $-2.88e^{-3}$ | $3.15e^{-4}$ | -9.12 | < 0.001 | 0.26 |
| Surface area per farmer above the forest line in 1851 | $3.22e^{-6}$ | $1.90e^{-6}$ | 1.69 | 0.094 | 0.01 |

On the other hand, the most parsimonious model accounted for a mere 6% of the total variation in the velocity of the forest-line shift between 1993 and 2010 and included only two variables (Table 3). The forest-line shift between 1993 and 2010 was faster with (1) a lower mean summer aridity index and (2) a greater decrease in livestock density between 2000 and 2010.

**Table 3: Estimates with associated standard errors, t-values and p-values for the linear model that predicting forest-line shift velocity between 1993 and 2010 in the French Pyrenees according to the mean summer aridity index in the 300 m above the 1993 forest line, and to change in livestock density between 2000 and 2010. The contribution of each variable to the total adjusted $R^2$ of 0.06 is presented in the last column.**

|  | Estimate | Std. Error | t value | p value | R² contribution |
|---|---|---|---|---|---|
| Total |  |  |  |  | 0.06 |
| Intercept | 20.55 | 7.40 | 2.78 | 0.006 | - |
| Mean summer aridity index | -32.45 | 12.63 | -2.57 | 0.012 | 0.05 |
| Change in livestock density | -0.74 | 0.37 | -1.99 | 0.049 | 0.02 |

For the closed forest line, four variables were kept in the most parsimonious model, which explained 28% of the total variation in the velocity of the shift between 1993 and 2010 (Table 4). The shift was faster with (1) a greater mean slope, (2) a lower proportion of forested area below the closed forest line, (3) a lower mean summer water balance (i.e. drier conditions) and (4) a lower closed forest-line elevation in 1993.

**Table 4: Estimates with associated standard errors, t-values and p-values for the linear model predicting closed forest-line shift velocity for the 1993-2010 period in the French Pyrenees based on mean summer water balance and mean slope in the 300 m above the 1993 closed forest line, proportion of forested area below the closed forest line, and closed forest-line elevation in 1993. The contribution of each variable to the total adjusted R² of 0.28 is presented in the last column.**

|  | Estimate | Std. Error | t value | p value | R² contribution |
|---|---|---|---|---|---|
| Total |  |  |  |  | 0.28 |
| Intercept | 6.63 | 4.47 | 1.48 | 0.141 | - |
| Mean summer water balance | -0.11 | 0.04 | -3.26 | 0.001 | 0.06 |
| Mean slope | 0.34 | 0.08 | 3.99 | < 0.001 | 0.10 |
| Proportion of forested area below the closed forest line | -8.55 | 2.53 | -3.37 | 0.001 | 0.07 |
| Closed forest-line elevation in 1993 | $-3.70e^{-3}$ | $1.79e^{-3}$ | -2.06 | 0.041 | 0.02 |

## 4 Discussion

### 4.1 Is the upward shift in the forest line occurring faster in the Pyrenees than in other mountain ranges in the Northern Hemisphere?

In consistence with the global trend (Harsch et al., 2009), we observed a clear upward shift in the forest line in the French Pyrenees. However, the average rate of the shift in the Northern Hemisphere between 1901 and 2018 for undisturbed limits (0.35 m.yr$^{-1}$; Lu et al., 2021) was four times less than the velocity of 1.44 m.yr$^{-1}$ we obtained for the same period in the French Pyrenees (Table S6). This suggests that forest lines in the Pyrenees were strongly limited by abiotic, biotic or human constraints before 1850 and that their release after this date resulted in the rapid upward shift we observed. Nevertheless, previous studies on the Pyrenees noted faster upward shifts compared to the average found for the Northern Hemisphere. A study in the French Catalan Pyrenees (the easternmost Pyrenees) found a rate of 0.7 m.yr$^{-1}$ between 1953 and 2015 (Feuillet et al., 2020). The

upward shift we found for the Catalan Pyrenees was one of the fastest in the whole study area. For the same period and the same municipalities in the Catalan Pyrenees, we estimated that the forest-line shift was faster still (1.9 m.yr$^{-1}$, Table S4, S6).

However, the methodologies used in the two studies were different: Feuillet et al. (2020) estimated forest-line shifts in 300 m radius buffers, while we estimated forest-line shifts at the larger scale of the municipality. This may explain the difference in magnitude for the shift rates. The municipality scale encompassed the full range of local conditions, thus compensating for sites with only a slow upward shift, or a downward shift, while the variability might not have been fully taken into account with Feuillet et al.'s small buffer zones. Similarly, the shift rate of 1.8 m.yr$^{-1}$ we found for the French Catalan Pyrenees (Table

S4, S6) was triple the rate of 0.7 m.yr$^{-1}$ reported between 1956 and 2006 by Améztegui et al. (2016) for the Spanish Catalan Pyrenees, located to the south of our study area. In this case, differences in the study locations (regional climate and soils) may be responsible for the slower rate observed in the Spanish Catalan Pyrenees, even though slope and exposure were similar to our study area.

Differences in forest-line shift rates could also be related to differences between recent and historical land-use maps or potential

deficiencies in the État-Major map. However, we believe that the sources used and the methods developed to vectorise this historical map and estimate forest-line elevations minimised these discrepancies. Firstly, the État-Major map is considered to be the most accurate document for working at the scale of an entire massif, and a good quality map for assessing forest cover in the 1850s (Dupouey et al., 2007). The État-Major map was based on the Napoleonic cadastre where available, which enhanced its accuracy (Rochel et al., 2017). Indeed, this was probably the case in the French Pyrenees, where the cadastral

data predated the État-Major map. Secondly, the standardisation of the maps that we have carried out should account for most of the differences between the products. Thirdly, definitions of the notion of "forest" could have differ between recent and historical land-use maps. Indeed, all areas were classified as forest on the EMM if the main income they produced came from forest products; areas were classified as "pasture" if the main income came from grazing (Abadie et al., 2018b; Rochel et al., 2017). In the absence of more precise information, a cautious approach would be to consider that the land designated as forest

on the EMM falls somewhere between the IGN definitions of 'forest' and 'closed forest' that we used in our study. If the forests on the EMM were considered to be closed forests, the shift rate between 1851 and 1993 would decrease strongly, to 0.8 m.yr$^{-1}$ for the French Catalan Pyrenees (Table S6). This suggests that the definition we applied to the forests on the EMM may have been partly responsible for the faster shift rates we detected. Fourthly, as for the État-Major map itself, a method of standardising vectorisation and georeferencing across France has been proposed and implemented in the French Pyrenees,

thereby reducing discrepancies between regions, particularly for vectorisation and georeferencing (Favre et al., 2017). This avoids the risk of introducing additional errors, given that the État-Major map was drawn by different people and there may be inconsistencies between map tiles in the cartographic figures (Dupouey et al., 2007; Thomas et al., 2017). Furthermore, forest boundary inaccuracies are more prevalent in mountainous areas and can be related to vectorisation and georeferencing (Thomas et al., 2017). Last but not least, using a forest cover threshold of 5% to determine the position of the forest line (see

2.4) reduces the risk of error for forest patches located at the top of municipalities, for example where scree is found today.

The new method developed in this study therefore contribute to reinforce the reliable estimations of forest-line elevation in historical maps and should be apply to other study areas in Europe and worldwide.

Moreover, going back to 1850 most probably allowed us to detect changes that occurred rapidly after the historic forest transition, as suggested by literature (Camarero and Gutiérrez, 2004) and indicated by the strong increase (+67%) in forest
area between 1851 and 1908 (Fig. 3b). Overall, this supports a faster shift in the forest line in the Pyrenees than in the rest of the Northern Hemisphere, which could be related to the high level of human activity in this mountain range.

**4.2 Is the forest line in the Pyrenees following the rise in the isotherm caused by global warming?**

Focusing on the temporal and regional warming trend, the Pic-du-Midi weather station has recorded increasing temperatures since the early 1900s, with a notable acceleration in recent decades. However, the raw temperature data suggests no temperature
warming between 1851 and 1910 (Fig. 3a). The fact that a significant temperature increase occurred only after 1900 in the French Pyrenees is supported by Marti et al. (2015). We could not check the daily temperatures to ascertain whether the conditions described by Paulsen and Körner (2014) are met for trees to be present: a growing season of at least 94 days, with a daily mean temperature of at least 0.9 °C and a mean temperature of 6.4 °C over all these days, as these data were not available for the whole studied period. Nevertheless, the warming of the mean temperature of the summer months (July,
August, September), when the mean temperature usually exceeds 6.4 °C on average, was observed to follow a similar trend as the mean annual temperature warming (Table S7). Moreover, we assumed that the warming experienced in the study area was homogeneous. This assumption is supported by Cuadrat et al. (2024), whose temperature extrapolation models for the period 1959–2000 revealed similar warming trends in the Atlantic and Mediterranean regions of the Pyrenees. Therefore, assuming no temperature change before 1910 and considering an adiabatic gradient of -0.55°C per 100 m elevation (Rolland, 2003), the
potential forest line should have shifted upward by 112 m from 1851 to 1993 and by 113 m from 1993 to 2010.

We observed an average forest-line shift velocity between 1851 and 1993 comparable to this theoretical shift (0.9 vs 0.8 m.yr$^{-1}$). This suggests that the effect of climate change on partly driving the forest-line dynamics could not be rejected. More than half of the municipalities (57%) had slower shift rates than the potential forest line, indicating spatial variations. Therefore, local estimations of potential forest line would be valuable to further estimate the difference between theoretical and observed forest-
line elevation (Shandra et al., 2013). However, such local estimation should rely on robust climatic past and present data covering the study period and area. Unfortunately, these climatic data are not available yet, and we think it would be more valuable to perform the whole analysis at a finer scale than the municipality.

Comparing the two studied periods, the acceleration in the upward shift we observed was concomitant to an acceleration in temperature increase (Fig. 3). Despite the uncertainties in comparing the two different periods, the role of temperature warming
in driving temporal forest-line dynamics is supported by a recent study, illustrating a forest-line shift acceleration in the Southern French Alps, driven by climate change (Nicoud et al., 2025). Indeed, the long period between the 1850s and the 1990s probably includes periods of stagnation and upward shifts, and the upward shift during such a long period is unlike to be linear. Conversely, the recent period between the 1990s and 2010 is very short. Despite the limited number of dates

available, we found it valuable to compare the periods in order to identify recent variations in forest-line dynamics trends.

Moreover, calculating shift velocities enabled us to make this comparison despite the different lengths of the two periods. Nevertheless, adding a supplementary date would be very valuable and may be possible in the near future with the preparation of a digital version of the 1950 French forest map.

Even with the acceleration, the average forest-line shift velocity of 3.5 m.yr$^{-1}$ we found between 1993 and 2010 is only half the theoretical rate (6.7 m.yr$^{-1}$). In addition, we observed an increase in the percentage of municipalities with slower shift rates

compared to the theoretical shift rate (78%). This discrepancy between theoretical and observed shift rates indicates that the forest-line shift is lagging behind climate warming (Beloiu et al., 2022; Körner and Hiltbrunner, 2024; Lloyd, 2005; Lu et al., 2021). The temporal lag in forest-line response can be related to the concept of climatic debt (Bertrand et al., 2016; Devictor et al., 2012). Considering the long life-cycle of trees and the time span of the forest development process, a considerable climatic debt is expected for forest shifts. However, we observed a shorter climatic debt for trees compared to what Richard et

al. (2021) detected for understory plants.

Despite the lag in response observed, the temporal dynamics of the upward forest-line shift in the French Pyrenees followed the rise in temperature over the last 160 years, suggesting that temperature has played an important role (Fig. 3a-b), as supported by previous literature (Hagedorn et al., 2014; Harsch et al., 2009). However, the spatial variations in forest-line shift rates we observed, and the absence of variables related to temperature in the most parsimonious linear mixed models indicate

other regional drivers may also be exerting a significant influence, in line with the findings of Améztegui et al. (2016) and Gehrig-Fasel et al. (2007).

### 4.3 Spatial patterns of forest-line shift rates are related to forest context

Our study emphasised for the first time that the forest-line shift rate varied strongly according to the tree species forming the forest line. Between 1851 and 1993, the forest lines in the east, dominated by the mountain pine, shifted faster than the forest

lines dominated by late-successional species, mainly found towards the west. Indeed, the mountain pine is a wind-dispersed species, able to colonise at some distance from its initial location, and it may more easily become established in the heath above the forest line (Camarero et al., 2005). Beech, on the contrary, colonises higher elevations slowly even when site conditions are favourable, and fir, although not limited by seed dispersal, is less competitive at the forest line, thus hindering its establishment (Axer et al., 2021; Scherrer et al., 2020). In a previous literature review, Hansson et al. (2021) found upward

or northward shifts in the forest line for 55% of the forest lines composed of angiosperms and for 72% of the forest lines composed of gymnosperms, though no clear effect of forest-line tree species was apparent. The authors propose that the over-representation of forest lines formed by Pinaceae and Betulaceae, in comparison to forest lines formed by other families, may have limited their ability to detect differences in forest-line shift rates among tree species. In the Pyrenees, our data included various groups of dominant tree species at the forest line, and we were therefore able to detect their effect on shift rate.

However, in BDF2, many of the pixels at the forest edge were classified as undifferentiated conifers and this somewhat limited our ability to determine precise species. In the subalpine Pyrenean forests we studied, however, mountain pine and fir were

the only two species present, so we were able to improve our species classification by including information from neighbouring municipalities and closed forest species. The "deciduous" class was probably more heterogeneous: it can include both late-successional species, for example beech, and early-successional species, for example mountain ash (*Sorbus aucuparia* L.) or

birch (*Betula sp*). Given that a number of species can collectively constitute the forest line in mountain ranges, it is of the utmost importance to accurately determine tree species composition (through, for example, a deep learning approach based on satellite data; Schwartz et al., 2023, 2024) in order to gain a deeper understanding of forest-line dynamics.

### 4.4 Is pastoral abandonment the main driver of the upward shift in forest line?

The shift between 1851 and 1993 was slower for higher forest-line elevations in 1851. However, the initial elevation had

almost no effect on the spatial variations in the forest-line and closed forest-line shift velocity between 1993 and 2010. This suggests that the initial forest-line elevation used to calculate the forest-line shift rate, had more than just a methodological link with the spatial distribution of forest-line shifts. Indeed, such methodological link should have existed over the two study periods. On other words, ecological factors are also likely to be involved. The initial forest-line elevation may in fact be an indicator of pastoral pressure before 1850, for which no pastoral data are available, nor spatially explicit climatic data to model

the potential forest-line at this time. It is likely that high previous pastoral pressure would have moved the forest line down to lower elevations, while less pastoral pressure would have allowed the forest line to follow its "natural" contour (Carreras et al., 1996; Ninot et al., 2008). Thus, a forest line displaced below its natural position by former intensive pastoral practices and located in an environmental context favourable for forest development would be susceptible to a faster upward shift in forest line once pastoral pressure was released (Améztegui et al., 2016). Conversely, if the forest line was near its highest natural

position initially, we would expect later upward shifts to be slower.

Historical pastoral surveys allowed us to document the early pastoral abandonment that occurred immediately following the forest transition of 1850, as suggested by population censuses (Fig. 3), and on a large scale (the entire French Pyrenees). The forest-line shift was faster where livestock density decreased the most: in the east between 1851 and 1993, and in the west between 1993 and 2010 (Table 3, Fig. 3, 6, S5). This pattern is consistent with the early abandonment of pastoralism in the

east and the persistence of pastoralism in the western part of the Pyrenees, before abandonment became widespread in the 2000s (Eychenne-Niggel, 2003; Métailié, 2006). Our results highlight the important role of pastoral abandonment on forest dynamics at the regional scale, rather than the more local scales investigated in previous European studies (Améztegui et al., 2016; Anselmetto et al., 2024; Gehrig-Fasel et al., 2007; Treml et al., 2016).

Furthermore, we observed a relation between changes in livestock density and dominant tree species at the forest line (Fig. S3).

Indeed, numerous transhumant sheep herds still remained in the western Pyrenees in the 1970s for milk and cheese production, while in the rest of the Pyrenees, a mixture of cattle and sheep for meat production became more common (Rinschede, 1977; Whited, 2018). Cattle consume less conifer foliage than do sheep; they should therefore have less impact on the colonisation of former pasture by mountain pine and fir (Wehn et al., 2011). Our results emphasise the importance of considering the

combined effects of pastoral abandonment and forest-line tree species composition in analyses of spatio-temporal variations in forest-line shift rates.

In addition, we found that mean summer water balance and aridity index had small but significant effects on forest-line shift rates (Table 2-4). We expected that the forest line would shift faster when droughts were shorter or less intense. However, our results indicated the opposite: the more favourable the water balance, the slower the forest line shifted upward. Moreover, we found a covariation for change in livestock density and climatic water balance along the longitudinal gradient of our study area (Fig. S3). Possibly, a more favourable water balance contributed to maintaining pastureland. Therefore, instead of a direct effect of climate heterogeneity on spatial variations in forest-line shift rates, pastoral abandonment may have played a prominent role. The spatial pattern was mainly influenced by forest context and pastoral abandonment dynamics, whereas the overall temporal dynamics mirrored climate warming. Our results represent a step forward in disentangling the effects of climate change from pastoral drivers of forest-line dynamics.

Even though our linear model for the 1993-2010 period included change in livestock density and the mean summer aridity index, it was only able to account for a small percentage of the variations in forest-line shift rate ($R^2 = 0.06$), contrary to the first period. This low percentage could be related to the absence of a consistent effect of dominant tree species class and initial forest-line elevation. These two variables were correlated with change in livestock density (Fig. S3), and there could have been an important delayed response to the accelerated pastoral abandonment in the 1950s (MacDonald et al., 2000). However, pastoral data were available only at the "arrondissement" scale and not at the "municipality" scale, making the data for changes in livestock density too coarse to reflect the complexities of recent pastoral abandonment and changes in practices. Socio-economic factors related to pastoralism should be further investigated at the municipality scale to validate the hypotheses arising from our results. Indeed, we made the analyses at the municipality scale to be able to link forest-line dynamics with socio-economic variables, and accounting for spatial variations across an entire massif. Because of anonymity issue, encountered when working on socio-economic data, in particular regarding farms in this study, socio-economic data are not available at a spatially explicit scale. For livestock data, the municipality scale was even too small and we had to use data at arrondissement level, which covers several municipalities. Nevertheless, working at this scale enabled us to combine data available at different spatial resolutions. Working at the municipality scale also had the advantage of shorten the calculation time. In addition, the second study period may have been too short (17 years vs. 142 years) to capture the link between these drivers and their impacts. The present study should be extended another decade, to lengthen the most recent period and better capture driver effects on recent forest-line dynamics. This is an interesting perspective for upcoming studies since the third version of the *BD Forêt*® is in preparation.

The closed forest-line shift was also faster in the west than in the east of the Pyrenees in the recent period. The main driver detected was slope, with steeper slopes leading to faster closed forest-line shift rates. This is consistent with the challenges of maintaining forestry and pastoral activities on steep slopes, leading to a faster closure of the forest (Abadie et al., 2018a). Contrary to our expectations, a lower proportion of closed forest in the municipality induced higher closed forest-line shift rates. This suggests that there was no "forest mass effect", but instead an effect of release from former pastoral pressure,

comparable to the effect associated with initial forest-line elevation. The negative effect of mean summer water balance also supports pastoral abandonment as a driver of closed forest-line shift, as it was for forest-line shift.

## 5 Conclusions

Thanks to our original approach involving historical forest maps over large spatial and temporal scales, we were able to document an early upward shift in the forest line in the French Pyrenees after the forest transition of the 1850s. We also highlighted the upward shift accelerated in recent decades. However, despite this acceleration, the forest line did not match the rate of regional temperature warming. The closed forest line moved upwards even faster than the forest line, emphasising a densification of the subalpine forest that may have implications for carbon sequestration. The shift in the forest-line was initially faster in the east, due to the early abandonment of pastoral activities and the dominance of mountain pine. However, following 1993, when pastoral abandonment became widespread, the shift accelerated faster in the west. Thus, drivers of the spatial patterns of forest-line shift rates (pastoral abandonment and forest context) differ from drivers of the temporal dynamics (climate). A more detailed information on the tree species forming the forest line is crucial to better understanding the patterns and drivers of forest-line dynamics. Finally, using municipalities as the units of analysis alongside a large spatial and temporal scale has proven to be an effective approach for examining the relationship between socio-economic factors and forest-line dynamics, and this methodology should be extended to other mountain ranges.

**Author contribution**

J-LD, CBKR and LB designed the study. CBKR and LB acquired the funding. NL collected geomatics data. J-LD, CBKR, LB and ND elaborated the processing of the data and the analyses. J-LD wrote the original script to process the data. ND conducted the data processing and analysis. SC, ET and ND collected and compiled additional socio-economic and forest data. All authors discussed the results. ND wrote the manuscript with the help of CBKR and LB. J-LD, SC and NL commented the manuscript.

**Competing interests**

The authors declare that they have no conflict of interest.

**Acknowledgements**

We would like to thank Aditya Acharya for his preliminary work on this topic as a Master's student. We are also grateful to Météo France, and in particular to Simon Gascoin and Jean-Michel Soubeyroux, for providing the temperature data from the Pic-du-Midi weather station. We thank Xavier Rochel, Jonathan Lenoir and Mélanie Saulnier for their advice.

**Financial support**

This work was supported by a grant overseen by the French National Research Agency (ANR) as part of the "Investissements d'Avenir" program (ANR-11-LABX-0002-01, Lab of Excellence ARBRE). The A2F pole at the University of Lorraine also contributed to funding this study. This work is part of the project MONITOR of the research program FORESTT and received government funding managed by the *Agence Nationale de la Recherche* under the *France 2030* program (reference ANR-24-PEFO-0003).

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
