# Peer review of "Long-term forest-line dynamics in the French Pyrenees: an accelerating upward shift related to forest context, global warming and pastoral abandonment"

_EGUsphere, 2024_

## Editor Comment (EC1)

**Main summary**

The manuscript is reconstructing the forest-line dynamics in the French Pyrenees and finds an upward shift of the tree-line which is related to both climate change and pastoral abandonment. The patterns of change are vastly different in the West and Eastern Pyrenees. The paper states two main goals: 1) the reconstruction of forest-line dynamics in the French Pyrenees from the minimum forest extent in 1850 to the present day, and 2) the exploitation of potential biophysical and anthropogenic drivers.

We note that this review document is jointly written by 1 senior researcher and 4 junior researchers. Each of us read and commented on the manuscript individually, then the paper was discussed as a group. We summarize here both the points we agree on as well as individual observations (marked as such).

We particularly enjoyed reading this manuscript and we think its main strengths lie in the fact that it reconstructs forest dynamics over very long time scales and a broad spatial area and that it relates these changes both to anthropogenic and climatic variation. The approach presented here is transferrable and while the reconstruction itself is not novel, there are many ingenious ways in which the authors deal with the limitations of the historical data (e.g. farming pressure, forest composition at tree line). We all felt that the research questions posed by the manuscript are answered, but we did have several major concerns that we feel need further clarification. Below, we describe these concerns and try to make suggestions on how to address them. We group our concerns in major concerns and other comments.

**Major concerns:**

The promise of climatic debt analyses and the tree line (not) tracking the isotherm shift is an overpromise of the paper that is not fully delivered on. These two concepts (climatic debt and isotherm shift) are mentioned only in abstract and discussion but not really analyzed per se in the text, albeit the fact that if they were, we feel like this paper would have a great(er) value. We believe that this conflict of the 'overpromising' manuscript could be solved by either 1) measuring actual climatic debt (eg. area that the forest could have moved to follow the isotherm, but did not) or by 2) not mentioning the climatic debt per se (because it is not really quantified here). The isotherm shift in our view would mean to find all points of a certain temperature (eg. 6 degrees at treeline) and spatially assess how much farther up those points are compared to historical baselines. We did not find this analysis described – but rather comparison to average change in temperature at a fixed point, which is interpreted as an upward isotherm shift.

The point above brings us to our second main concern: we did not fully grasp how the climatic (claimed isotherm shift at forest line) can be inferred from weather data that was sourced from a single weather station at Pic-Du-Midi which is located nearly 1000m higher than the forest line. The use of the climatic data was not clear to us from the text,but beyond this clarification: if indeed data from a single location was used for the entire study region, we find this problematic because a) it wipes out local variability. If spatially explicit climate data was indeed used, then the description of how this data was used and calculated needs to be clearer and offer similar weight and space in the manuscript as the pastoral pressure data.

We all agreed that while it is great that the study covers such an expanded timescale, there needs to be some more discussion and depending on possibilities also a sensitivity analyses that account for the way in which measuring annual change between only two points in time, may confound 'noise' with actual trends.

**Other comments:**

I would welcome a more thorough discussion of why the data was analyzed at municipality level rather than not used spatially explicitly. I believe this may be because of the anthropogenic variables, but then again the authors did a great job spatializing some of those too (eg. by accounting for livestock owners/ pasture).

L100: no overarching goal/ objective for the manuscript is stated – in addition to the specific questions.

Table 1: with variables could be accompanied by another column with expected effect direction
Figure 3b: the right axis scale seems to be transformed because values are not equidistant – can you detail the transformation applied?
L190: how was the data from Pic-du-Midi extrapolated to the whole study region? And if it was not, how can the link be made between isotherm shifts at more than 1000 m below this elevation?

We found that in parts the text provides excessive detail (eg. the map reconstruction, variables regarding grazing) and in others the text was particularly low in detail. For example, processing of the variables in cases where variables were not used should not be described in detail in the text but rather only in the supplement. Some examples include but are not limited to:

> L 220: why was the map from 1908 not used in the analyses of forest line change (it would have been great for mitigating the issue of variability within the two very distant observations from 1851 and 1993) but then mentioned in the text here?

> L280: why is the description of grazing intensity relevant here

Line 280: need to state how models were selected (forward, backward selection). Please list the final best selected model for each section.

L350: the forest line velocity shift model for 1993-2010 and for closed forest is explaining a very low amount of variation. I would deem this a bad model – is there an explanation why this is considered acceptable? What else may be missed in terms of variables?

L123: In the chapter 2.2 Selection of the studied municipalities you say that you exclude the municipalities that had already reached the maximum elevation of forest line in 1993 or 2010. Why do you exclude those from the study and would those municipalities not also be in line with the first objective of the study?

L160: Which forest extent is represented in the newer maps? the open forest line or closed forest line? In the sentence before the figures you mention that you create two vector maps for BDF1 and BDF2. Maybe you do not see a difference at that scale but it seems strange displaying three maps, when 5 are mentioned beforehand without specification which ones they are.

L262 - 283: You describe the variables for the three models. Maybe a graph with the same information would be more intuitive.

L396-402: Because you only mention in line 209 in the description of the graph that you use the temperature shift to estimate the upward shift in forest line and there is no further explanation in the methods, this part is confusing. You should explain that you use the + 0,62°C temperature shift in 1910 - 1993 and + 0,62°C for 1993 - 2010 in combination with the adiabatic gradient of - 0,55 °C / 100 m elevation to calculate the respective 112m theoretical upward shift in one place (considering the concerns about this broad approach earlier).

L331: In figure 6 the timespan mentions 1994 several times instead of 1993. The figure title also says 1993.

L166 and L172: metre instead of meter.

References: this piece of work from Eastern Europe on similar time scales and topics may be relevant: https://link.springer.com/chapter/10.1007/978-3-642-12725-0_16

**Review Questions:**

1. Does the paper address relevant scientific questions within the scope of BG

yes

2. Does the paper present novel concepts, ideas, tools, or data?

yes

3. Are substantial conclusions reached?

Yes, potential for it, but see major concerns above

4. Are the scientific methods and assumptions valid and clearly outlined?

The climatic data analyses needs clarification. More clarity on forest context metrics is needed.

5.      Are the results sufficient to support the interpretations and conclusions?

Partly

6.      Is the description of experiments and calculations sufficiently complete and precise to allow their reproduction by fellow scientists (traceability of results)?

At times too detailed, at times too little detail. See 'other comments'. Further methodological details would help.

7.      Do the authors give proper credit to related work and clearly indicate their own new/original contribution?

We did not check all references in detail!

8.      Does the title clearly reflect the contents of the paper?

Yes

9.      Does the abstract provide a concise and complete summary?

The abstract feels a bit like an overstatement, it overpromises and the reader is bound to be disappointed in the actual paper. I think the abstract could be 'toned down' a bit.

10.      Is the overall presentation well structured and clear?

Yes

11.      Is the language fluent and precise?

Language could be checked and simplified. See also Q6.

12.      Are mathematical formulae, symbols, abbreviations, and units correctly defined and used?

NA

13.      Should any parts of the paper (text, formulae, figures, tables) be clarified, reduced, combined, or eliminated?
See major comments

14.      Are the number and quality of references appropriate?

Yes

15.      Is the amount and quality of supplementary material appropriate?

Yes

---

## Author Response (AR1)

**Answer to RC1**: 'Comment on egusphere-2024-4099', Anonymous Referee #1

Thank you for your positive feedback and the constructive comments. You will find a response to each part below, including the changes we have proposed based on your comments.

INTRODUCTION

We have reorganised the introduction as you suggested. We began with the generalities of treeline position, then moved on to the case of the Pyrenees, addressed the knowledge gap, and finished by considering the benefits of using historical maps.

At the scale of the French Pyrenees, vectorised maps of forest cover were only available for the three dates we mentioned. An ongoing project aims to vectorise forest land use from the 1950s map and will be used in a near future in the French Alps, but is not yet available for the French Pyrenees. Concerning the most recent forest cover map, the next edition of the French forest map (BD Forêt® v3) is actually scheduled but for 2027 only.

METHODOLOGY:

We have improved the readability of this section, as you suggested.

Of the 25 m, 50 m and 100 m elevational bands that we tested, the 100 m band showed the most consistent distribution of forest cover according to elevation with the least background noise (Figure 2). Consequently, the 100 m elevation band was the most appropriate for estimating the elevation of the limit at a canopy cover threshold in the next step of the method. We have mentioned it in the main text in the revised version.

In our manuscript, we distinguished between temporal and spatial analyses. To assess the temporal variations in temperature, only data from the Pic-du-Midi weather station allowed us to go back to the 19th century. The other weather stations in the Pyrenees had shorter time series and were located at lower elevations. To assess the spatial variations, we focused on recent data and used the Aurelhy database to extract climatic data. We tested the derived variables "mean temperature in the warmest month of summer", "mean temperature in the coldest month of winter" and "total annual precipitation" for the period 1981-2010 and in the 300 m above the forest line (in 1851 or in 1993) in the linear models, but they were rarely retained in the most parsimonious models. During preliminary analyses, we also tested temperature changes between the periods 1961-1985 and 1986-2010 in each municipality, but these were not retained in the models either. Furthermore, when we compared temperature changes between the periods 1961-1990 and 1981-2010 from the homogenised series (seven stations in the study area) with those from the climatic models (Aurelhy, SAFRAN, CHELSA v2.1 and CHELSAcruts), we found no correlation. This shows that the spatial models are unable to accurately characterise temperature change in the study area of the French Pyrenees. We have clarified this in the new version.

Based on the data we have, we cannot assess the proportion of livestock grazing in the mountains or whether farming is extensive or intensive. However, we can distinguish between sheep and cattle. After investigation, we observed a decrease in the number of sheep and a slight increase in the number of cattle. We have added this detailed temporal trend to the new version of the manuscript. However, including change in livestock density for cattle and sheep rather than change in total livestock density did not significantly improve the models explaining spatial variations. Therefore, we kept the previous variables and models.

RESULTS, MAPS, TABLES AND FIGURES

Figure 1: We added the elevation from the digital elevation model as a background and used a light blue colour for the ocean. We also added a scale to panel A.

We enhanced the visual quality of the figures. Unfortunately, converting them to PDF seems to have degraded them again.

Table 1: The range has been removed from the table in the main text.

Figure 4: The position of the figure caption has been adjusted to provide a clearer indication of what the colours correspond to.

Figure 5: The x-axis has been adjusted and the boxplots have been displayed.

Figure 6: As for Figure 1, a background has been added. A color-blind-friendly palette has been used.

DISCUSSION

In the preprint, we argued that the higher rate of change observed in this study compared to previous studies is probably related to differences in the study area or methodology when working in the same area. We didn't address the potential deficiencies of the État-Major map, as it is considered to be the most accurate document for working at the scale of an entire massif, and a good quality map for assessing forest cover in the 1850s (Dupouey et al., 2007). Moreover, the État-Major map was based on the Napoleonic cadastre where available, which enhanced its accuracy (Rochel et al., 2017). Indeed, this was probably the case in the French Pyrenees, where the cadastral data predated the État-Major map. Verifying the correspondence between the Napoleonic cadastre and the État-Major map would be valuable for completing the historical data, but this is a complex task for such a regional study area. Nevertheless, we agree that some of the discrepancy may be due to differences between the maps or potential deficiencies in the État-Major map. However, we believe that the methods developed to vectorise the historical map and estimate forest-line elevations minimised these discrepancies. Firstly, the standardisation of the maps that we have carried out should account for most of the differences between the products. Secondly, as we have already discussed, the definition has limitations, as the forest on the État-Major map probably falls somewhere between the IGN definitions of "forest" and "closed forest". Thirdly, as for the État-Major map itself, a method of standardising vectorisation and georeferencing across France has been proposed and implemented in the French Pyrenees, thereby reducing discrepancies between regions, particularly for vectorisation and georeferencing (Favre et al., 2017). Indeed, the État-Major map was drawn by different people, so there may be inconsistencies between map tiles in the cartographic figures (Dupouey et al., 2007; Thomas et al., 2017). Furthermore, forest boundary inaccuracies are more prevalent in mountainous areas and can be related to vectorisation and georeferencing (Thomas et al., 2017). Last but not least, using a forest cover

threshold to determine the position of the forest line reduces the risk of error for forest patches located at the top of municipalities, for example where scree is found today. We have developed these considerations in the new version of the manuscript, as you suggested.

References:

Dupouey, J.-L., Bachacou, J., Cosserat-Mangeot, R., Aberdam, S., Vallauri, D., Chappart, G., and Corvisier-de-Villèle, M.-A.: Vers la réalisation d'une carte géoréférencée des forêts anciennes de France, Le Monde des Cartes, Paris: Comité français de cartographie, 85–98, 2007.

Favre, C., Grel, A., Granier, E., Cosserat-Mangeot, R., Bachacou, J., Leroy, N., and Dupouey, J.-L.: Digitalisation des cartes anciennes. Manuel pour la vectorisation de l'usage des sols et le géoréférencement des minutes 1:40 000 de la carte d'Etat-Major. Version 13.3, INRA, 79, 2017.

Rochel, X., Abadie, J., Avon, C., Bergès, L., Chauchard, S., Defever, S., Grel, A., Jeanmonod, J., Leroy, N., and Dupouey, J.-L.: Quelles sources cartographiques pour la définition des usages anciens du sol en France ?, Rev. For. Fr., 353, https://doi.org/10.4267/2042/67866, 2017.

Thomas, M., Bec, R., Abadie, J., Avon, C., Bergès, L., Grel, A., and Dupouey, J.-L.: Changements à long terme des paysages forestiers dans cinq parcs nationaux métropolitains et le futur parc national des forêts de Champagne et Bourgogne, Rev. For. Fr., 387, https://doi.org/10.4267/2042/67868, 2017.

SPECIFIC COMMENTS

We have incorporated all of the following answers in the new version of the manuscript.

L36: These studies have been conducted in Europe, more precisely in the Alps and Pyrenees, for these examples.

L41-44: I propose changing the second sentence to make it clearer as follows: "Recent upward shifts in the forest line have also been reported at large spatial scales thanks to aerial photographs (Améztegui et al., 2016; Gehrig-Fasel et al., 2007). However, only a limited response of the forest line, or a lag in the response, has been documented in **several locations across Europe** despite increasing temperatures (Gehrig-Fasel et al., 2007; Körner and Hiltbrunner, 2024; Paulsen et al., 2000)."

L49-50: Across Europe, historical maps can cover large areas, as the État-Major map (Kaim et al., 2016). However, the use of historical maps was usually restricted to small areas in comparison to entire massifs (eg. Egarter Vigl et al., 2016; Mainieri et al., 2020; Mietkiewicz et al., 2017; Tasser et al., 2007).

References:

Egarter Vigl, L., Schirpke, U., Tasser, E., and Tappeiner, U.: Linking long-term landscape dynamics to the multiple interactions among ecosystem services in the European Alps, Landscape Ecol, 31, 1903–1918, https://doi.org/10.1007/s10980-016-0389-3, 2016.

Kaim, D., Kozak, J., Kolecka, N., Ziółkowska, E., Ostafin, K., Ostapowicz, K., Gimmi, U., Munteanu, C., and Radeloff, V. C.: Broad scale forest cover reconstruction from historical topographic maps, Applied Geography, 67, 39–48, https://doi.org/10.1016/j.apgeog.2015.12.003, 2016.

Mainieri, R., Favillier, A., Lopez-Saez, J., Eckert, N., Zgheib, T., Morel, P., Saulnier, M., Peiry, J.-L., Stoffel, M., and Corona, C.: Impacts of land-cover changes on snow avalanche activity in the French Alps, Anthropocene, 30, 100244, https://doi.org/10.1016/j.ancene.2020.100244, 2020.

Mietkiewicz, N., Kulakowski, D., Rogan, J., and Bebi, P.: Long-term change in sub-alpine forest cover, tree line and species composition in the Swiss Alps, J Vegetation Science, 28, 951–964, https://doi.org/10.1111/jvs.12561, 2017.

Tasser, E., Walde, J., Tappeiner, U., Teutsch, A., and Noggler, W.: Land-use changes and natural reforestation in the Eastern Central Alps, Agriculture, Ecosystems & Environment, 118, 115–129, https://doi.org/10.1016/j.agee.2006.05.004, 2007.

L111: "The Pyrenees range stretches over 300 km between the Atlantic Ocean and the Mediterranean Sea, **at the French-Spanish border**, and is almost 100 km wide in its central part."

L116-118: The values are for the French Pyrenees. This have been indicated. "The eastern region is under Mediterranean influence and is characterised by lower precipitation and a warmer average temperature than the western region, under oceanic influence: 1060 vs 2298 mm of average annual precipitation and 5.9 vs 5.3°C between 1958 and 2008, respectively for Cerdagne (eastern **French** region) and the Pays-Basque massifs (western **French** region) (Maris et al., 2009)."

L143-144: We did not have information on forest cover in the other countries. We believe that such information would have been valuable without altering the results. Indeed, I could not find a suitable forest map in Spain dating from the 19th century (Ruiz Del Castillo Y Navascues et al., 2006). Furthermore, in most cases, the boundary follows the ridge. Therefore, the probability that a small patch extend in the neighbouring countries is reduced, as the alpine belt is often reached. Moreover, in the case where the forest continued at higher elevation in the neighbouring country, the forest most likely reached the highest elevation in the municipality, this municipality was thus excluded from the analyses.

Reference:

Ruiz Del Castillo Y Navascues, J., López Leiva, C., García Viñas, J. I., Villares Muyo, J. M., Tostado Rivera, P., and García Rodríguez, C.: The Forest Map of Spain 1:200,000. Methodology and analysis of general results, For. syst., 15, S24–S39, https://doi.org/10.5424/srf/200615S1-00979, 2006.

L167: "For each municipality in the study area, we rasterised the forest **present** in each vector **map** (Fig. 2a)."

L222: The Daubrée forest inventory only reports forest area for each municipality but this historical source is not at all a land-use map. Therefore, we only use it to estimate changes in forest cover for the entire study area. Nevertheless, we have properly cited this source of information in M&M.

L247: Including livestock density data from 1838 allows us to visualise changes in the temporal trend in the study area since the 1850s: an increase before 1850, followed by a decrease. Changes in livestock density between 1852 and 2000 are used in the models for this period to assess spatial variations in the study area. Here again, the distinction between temporal and spatial analysis have been clarified.

L255: The models were built with the *lme* function of the R package *nlme* (Pinheiro et al., 1999). The response variables were (1) forest-line shift velocity (between 1851 and 1993, between 1993 and 2010 and between 1851 and 2010), (2) the differences in forest-line shift velocity between the two periods (1851-1993 and 1993-2010), (3) the differences in shift velocity between the forest line and closed forest line, therefore accounting for municipality pairing. These three models were built using only the "arrondissement" as a random effect, with no fixed effects, to test if the differences were significant. While they are similar to t-tests, they allowed us to account for the "arrondissement" effect, i.e. spatial autocorrelation. This was probably unclear in the first preprint, but we have clarified this in the new version. For the last test (4) the forest-line shift velocity between 1851 and 1993 and between 1993 and 2010 were the response variables and the effect of the spatial group was tested (west vs east, separated at an RGF93 longitude of 520 km).

Reference:

Pinheiro, J., Bates, D., and R Core Team: nlme: Linear and Nonlinear Mixed Effects Models, https://doi.org/10.32614/CRAN.package.nlme, 1999.

L284: I chose the threshold of 1.75 to simplify the selection of the most parsimonious models, by comparing fewer extracted models while retaining those with only small variations in the adjusted $R^2$ and with at least two variables. We have finally used a threshold of 2 in the revised manuscript and adjusted the results accordingly.

L355: A lower mean summer water balance means a lower quantity of water available for plants, after accounting for evaporation and transpiration, i.e. drier conditions. We have specified this in the new version of the manuscript.

L400-408: The data I used are monthly mean temperatures. Therefore, we cannot check the daily temperatures to ascertain whether the conditions described by Paulsen and Körner (2014) are met: a growing season of at least 94 days, with a daily mean temperature of at least 0.9 °C and a mean temperature of 6.4 °C over all these days. Still, we can calculate the mean temperature for the summer months (July, August, September), when the mean temperature usually exceeds 6.4 °C on average. This produces a similar trend to that of the annual mean temperature: an increase in growing season temperatures of 0.75°C between 1851 and 1993, resulting in a theoretical shift in the forest line of 136 m (and a velocity of 1.0 m.yr$^{-1}$); and an increase of 0.70°C between 1993 and 2010, resulting in a theoretical shift of 126 m (and a velocity of 7.4 m.yr$^{-1}$).

Reference:

Paulsen, J. and Körner, C.: A climate-based model to predict potential treeline position around the globe, Alp Botany, 124, 1–12, https://doi.org/10.1007/s00035-014-0124-0, 2014.

**Answer to RC2: 'Comment on egusphere-2024-4099', Anonymous Referee #2**

Thank you for your review and valuable comments. We have emphasise the novelty of the methodology in the discussion section of the new version of the manuscript.

Please, find a detailed answer to each point below.

We are aware that the two periods under comparison are quite different. As you pointed out, the long period between the 1850s and the 1990s probably includes periods of stagnation and upward shifts, and the upward shift during such a long period is unlike to be linear. Despite the limiting availability of dates, we found it valuable to compare the periods in order to identify recent variations in forest-line dynamics trends. Moreover, calculating shift velocities enabled us to make this comparison despite the different lengths of the two periods. Furthermore, previous studies have found evidence of an upward shift in forest line in the Spanish Pyrenees early after the 1850s (Camarero and Gutiérrez, 2004). As land was abandoned early in the French Pyrenees, at least on the eastern side, the forest line could have started to shift upward early too. However, studies in the Alps found later forest-line upward shifts, starting in the 1950s, and this pattern may also be observed in the French Pyrenees. Adding a supplementary date would be very valuable and may be possible in the near future with the preparation of a digital version of the 1950 French forest map. Conversely, the recent period between the 1990s and 2010 is very short. Nevertheless, we were able to capture forest-line dynamics and spatial variations. Therefore, we believe that these two periods are still relevant to this study, and that the comparison is valuable when all of this is taken into account. We have addressed these issues and the limitations of this comparison in the new version of the manuscript.

Reference:

Camarero, J. J. and Gutiérrez, E.: Pace and pattern of recent treeline dynamics: response of ecotones to climatic variability in the Spanish Pyrenees, Climatic Change, 63, 181–200, https://doi.org/10.1023/B:CLIM.0000018507.71343.46, 2004.

The original État-Major map has a scale of 1:40,000. BD Forêt® V1 and V2 were created from the interpretation of aerial photographs at scales ranging from 1:17,000 to 1:25,000. We have mentioned this in the new version of the manuscript. As you underlined, the minimum mapping unit differs between the three maps. That's why, in a first step, we have downgraded the accuracy of the État-Major map and BD Forêt® v2 at the level of the BD Forêt® v1. Thus, these differences should not impact the estimations of forest-line dynamics. We have also clarified this point.

The Daubrée forest inventory data only included forest area, with no map or precise spatial indications. This prevented us from mapping forest cover and estimating forest-line elevation at this date. However, we believe that using the forest area information for comparison is still interesting. Figure 3 therefore shows four points for forest area (light green triangles) and three for forest-line elevation (dark green points). We have clarified the presentation of this additional data source and the figure 3 in the new version of the manuscript.

During the preliminary analyses, we examined the maximum forest elevation per municipality, which should correspond to your proposal for the position of the local maximum, albeit at a larger scale. However, the maximum forest elevation was more susceptible to potential mapping errors. Moreover, although present, the differences between the two limits (5% forest cover in the highest elevation band of 100 m and maximum forest limit) were still quite small in most municipalities, thus insufficient to exclude the impact of human activity. Therefore, we don't think this additional analysis will be relevant in the case of the French Pyrenees.

Potential treeline elevation modelling relies on climatic data. While it is possible to do this for recent times, spatially consistent climatic reconstructions going back to the 1850s are lacking. This prevents us from estimating the deviation from the potential treeline elevation at the beginning of the study period. As Ameztégui et al. (2016) previously proposed using forest-line elevation as a proxy for human impact, and this allowed quantification even in the 1850s, we have chosen to use a similar indicator in our study. Furthermore, we believe that estimating the potential treeline elevation would be more relevant at a finer scale than for the entire municipality, although we agree on the usefulness of this approach for answering these questions.

Reference:

Améztegui, A., Coll, L., Brotons, L., and Ninot, J. M.: Land-use legacies rather than climate change are driving the recent upward shift of the mountain tree line in the Pyrenees, Global Ecology and Biogeography, 25, 263–273, https://doi.org/10.1111/geb.12407, 2016.

The relevant references you proposed to add (DOI: 10.1657/AAAR0013-108 ; https://doi.org/10.1111/jvs.12448) have been cited in the new version of the manuscript.

Minor comments

L62: We used the term 'potential treeline' to clarify the target object, considering the possibility of a 'realised treeline'. This is because the treeline is not considered a potential line as a consensus in the literature.

L201: We chose 300 m because it is small enough to be representative of conditions where forest-line shifts should occur, but large enough to include enough points for a meaningful average.

L265: We also checked the VIF, which resulted in a similar selection of variables. However, comparing the correlations enabled us to test several variable combinations in the preliminary analysis. We therefore continued with this approach for the final set selection.

Chapter 4.2: This was not so evident to us from the literature, as the effects of climate and land-use changes probably interact. Evidence of the impact of both climate change (Hagedorn et al., 2014;

Harsch et al., 2009) and land-use change (Améztegui et al., 2016; Gehrig-Fasel et al., 2007) on forest-line dynamics has previously been observed. Moreover, whether or not it is a coincidence, the similar pace of forest-line shift and temperature increase does not allow us to reject the hypothesis that climate change is a driver. Conversely, the fact that the forest line is shifting more rapidly than the temperature is warming suggests that climate change is not the only driver. We have clarified this idea in the revised version of the manuscript.

References:

Améztegui, A., Coll, L., Brotons, L., and Ninot, J. M.: Land-use legacies rather than climate change are driving the recent upward shift of the mountain tree line in the Pyrenees, Global Ecology and Biogeography, 25, 263–273, https://doi.org/10.1111/geb.12407, 2016.

Gehrig-Fasel, J., Guisan, A., and Zimmermann, N. E.: Tree line shifts in the Swiss Alps: climate change or land abandonment?, Journal of vegetation science, 18, 571–582, 2007.

Hagedorn, F., Shiyatov, S. G., Mazepa, V. S., Devi, N. M., Grigor'ev, A. A., Bartysh, A. A., Fomin, V. V., Kapralov, D. S., Terent'ev, M., Bugman, H., Rigling, A., and Moiseev, P. A.: Treeline advances along the Urals mountain range – driven by improved winter conditions?, Global Change Biology, 20, 3530–3543, https://doi.org/10.1111/gcb.12613, 2014.

Harsch, M. A., Hulme, P. E., McGlone, M. S., and Duncan, R. P.: Are treelines advancing? A global meta-analysis of treeline response to climate warming, Ecology Letters, 12, 1040–1049, https://doi.org/10.1111/j.1461-0248.2009.01355.x, 2009.

**Answer to RC3**
We thank everyone who took part in this review for their feedback. We have addressed the various concerns raised in the following text.

Major concerns:

In the new version of the manuscript, we have not mentioned climatic debt or the lag of forest line shift behind temperature warming in the abstract. However, we have still discussed these topics, as we believe them to be valuable despite the limitation of using data from only one weather station at a high elevation. Indeed, the Pic-du-Midi weather station is the only one in the study area that allows us to go back as far as the 1880s. Other stations have only collected recent data and are located far below the forest line. In the manuscript, we proposed two analyses: one focusing on the temporal trend and the other on the spatial variations. Regarding the temporal analysis, we focused on time depth and considered that the regional trend in temperature change was consistent across the entire study area. We assumed the data from the Pic-du-Midi weather station was representative of this trend. We acknowledge that local discrepancies in climate may have occurred and addressed this question by analysing the relationship between spatial variations in forest-line shift and present climatic data. Indeed, for the analysis of spatial variations, we used mean temperatures in the warmest month of summer and in the coldest month of winter, calculated for the period 1981-2010 from the Aurelhy database (see Chapter 2.5.1 of the manuscript). Furthermore, an analysis of spatially explicit differences in relation to the potential forest line would be more meaningful at a finer spatial scale than the municipality scale. Please note that the main reason we applied our analysis at the municipality scale was that this spatial scale was the appropriate scale to investigate the effect of socio-economic variables. We have clarified this in the new version of the manuscript.

We have added a discussion point on using annual rates to compare the two periods in the manuscript. Annual rates are indeed necessary when comparing periods of different lengths. However, rather than measuring annual changes directly, we applied regression slopes to the entire period. Furthermore, rather than reflecting annual changes, the standard error illustrated spatial variations, occurring among municipalities. Nevertheless, there are limitations as we only used three dates. For example, we assumed that the minimum extent of the forest was around the middle of the 19th century, when the État-Major map was drawn. However, in some departments, including municipalities inside the study area (Haute-Garonne and Pyrénées-Atlantiques), the minimum extent of the forest was probably later, around the beginning of the 20th century. The forest line may therefore have shifted downwards first and upwards at a greater speed afterwards. Nevertheless, the three dates we used were the only ones for which we had data on the entire study area at a fine scale. For more information on this issue, please refer to the response to RC2 (https://doi.org/10.5194/egusphere-2024-4099-AC2).

Other comments:
We conducted the analyses at the municipality level to link forest-line dynamics with socio-economic variables. Using the example you mentioned, the number of farmers is not available at a lower resolution than the municipality level, which is already sometimes too small to avoid anonymity issues. The same was true of the livestock data, and we had to use data at arrondissement level, which covers several municipalities. Nevertheless, working at this scale enabled us to combine data available at different spatial resolutions. Furthermore, conducting a spatially explicit analysis would have resulted in longer calculation times for such a large study area. We have discussed this in the new version of the manuscript.

L100: A more general objective have been included in the new version of the manuscript.

Table 1: The first reviewer argued that this table already contained too much information. But, we have tried to address both issues and proposed a new version of the table in the second manuscript version.

Figure 3b: Indeed, the right axis has been log-transformed to keep a consistent scale in comparison to the forest-line shift data. This have been mentioned in the new version of the manuscript.

L190: The data were not extrapolated to cover the whole study area. Despite being located above the forest line, the Pic-du-Midi weather station was the only weather station in the study area of the Pyrenees that provided data dating back to the 1880s at a high elevation. Without any other historical data, we therefore worked with anomalies and assumed that the warming experienced in the study area was homogeneous, as supported by Cuadrat et al. (2024).

Reference:
Cuadrat, J. M., Serrano-Notivoli, R., Prohom, M., Cunillera, J., Tejedor, E., Saz, M. Á., De Luis, M., Llabrés-Brustenga, A., and Soubeyroux, J.-M.: Climate of the Pyrenees: Extremes indices and long-term trends, Science of The Total Environment, 933, 173052, https://doi.org/10.1016/j.scitotenv.2024.173052, 2024.

L220: The Daubrée forest inventory of 1908 only provided tables on forest surface area per municipality but did not map the forest patches in 1908. In this part of the analysis, we focused on the temporal change in the forest surface area for the whole region to document this global trend on figure 3b. We have clarified this in the new version of the manuscript.

L280: The next sentence regarding missing values for the surface area per farmer variable have been moved to the supplementary material.

Line 280: The models were selected using the *dredge* function in the *MuMIn* package This model selection procedure is different from a backward or a forward selection because its objective is to compare all together the possible models including 1, 2, ..., n predictors; please refer to the presentation of the dredge function (Bartoń, 2023). It returns a table containing the different models, the list of predictors for each model and the associated characteristics (R², adjusted R², degrees of freedom, AICc, etc.). To select the best model, we extracted all the models with delta(AICc)<2 compared to the model with the lowest AICc, and selected the model with the fewest variables (i.e. the most parsimonious one). This have been clarified and the final best model have been listed for each section in the new version of the manuscript.

Reference:
Bartoń, K.: Package 'MuMIn,' 2023.

L350: We agree that the models explained a very low amount of variation, in particular for the forest-line shift velocity between 1993 and 2010. However, despite testing a lot of different variables, we were not able to find better models. The short time period considered may be responsible for the lack of explanation, as may the spatial resolution of the explanatory variables available to us.

L123: We excluded municipalities where the forest had already reached the highest elevation, as it is not possible to observe a forest-line upward shift in these municipalities. Including them would bias the trend estimate. Furthermore, we verified that no downward shift had occurred in these municipalities.

L160: The forest line is represented. We used the term 'forest line' to refer to all forests, including both open and closed ones. We only compared closed forest lines for the recent period and believe that including five lines/maps in the method figure would have made it less clear. However, this have been clarified in the new version of the manuscript.

L262 - 283: Regarding the description of the variables for the models, we already have a table in the main text and a figure in the supplementary material which we believe help to summarise the written information. We have added information about which variables are included in the models to Table 1, and we have moved the description of these variables from the main text to the supplementary material.

L396-402: The graph only aimed to show the temperature shift. I don't think it is relevant to discuss the adiabatic gradient in this figure, especially since the theoretical shift is not represented. I would therefore prefer to remove this information. Furthermore, this would be consistent with only mentioning it in the discussion section.

L331: This error have been corrected in the new version of the manuscript.

L166 and L172: The text was written in British English, so 'metre' is the correct spelling. Furthermore, the first version submitted has been revised by a native British speaker.

References:
Thank you for the relevant reference (https://link.springer.com/chapter/10.1007/978-3-642-12725-0_16), which have been included in the new version of the manuscript.